# Long noncoding RNA LncHIFCAR/MIR31HG is a HIF-1α co-activator driving oral cancer progression

Jing-Wen Shih[1,2,3,4], Wei-Fan Chiang[5,6], Alexander T.H. Wu[3,7], Ming-Heng Wu[3], Ling-Yu Wang[4], Yen-Ling Yu[8], Yu-Wen Hung[8], Wen-Chang Wang[3], Cheng-Ying Chu[2], Chiu-Lien Hung[4], Chun A. Changou[1,3,4,9], Yun Yen[1] & Hsing-Jien Kung[1,2,4,8]

Long noncoding RNAs (lncRNAs) have been implicated in hypoxia/HIF-1-associated cancer progression through largely unknown mechanisms. Here we identify MIR31HG as a hypoxia-inducible lncRNA and therefore we name it LncHIFCAR (long noncoding HIF-1α co-activating RNA); we describe its oncogenic role as a HIF-1α co-activator that regulates the HIF-1 transcriptional network, crucial for cancer development. Extensive analyses of clinical data indicate LncHIFCAR level is substantially upregulated in oral carcinoma, significantly associated with poor clinical outcomes and representing an independent prognostic predictor. Overexpression of LncHIFCAR induces pseudo-hypoxic gene signature, whereas knockdown of LncHIFCAR impairs the hypoxia-induced HIF-1α transactivation, sphere-forming ability, metabolic shift and metastatic potential in vitro and in vivo. Mechanistically, LncHIFCAR forms a complex with HIF-1α via direct binding and facilitates the recruitment of HIF-1α and p300 cofactor to the target promoters. Our results uncover an lncRNA-mediated mechanism for HIF-1 activation and establish the clinical values of LncHIFCAR in prognosis and potential therapeutic strategy for oral carcinoma.

[1] Ph.D. Program for Cancer Biology and Drug Discovery, College of Medical Science and Technology, Taipei Medical University, Taipei 110, Taiwan. [2] Research Center of Cancer Translational Medicine, Taipei Medical University, Taipei 110, Taiwan. [3] Ph.D. Program for Translational Medicine, College of Medical Science and Technology, Taipei Medical University, Taipei 110, Taiwan. [4] Department of Biochemistry and Molecular Medicine, Comprehensive Cancer Center, University of California at Davis, Sacramento, CA 95817, USA. [5] Department of Oral and Maxillofacial Surgery, Chi-Mei Medical Center, Liouying, Tainan 736, Taiwan. [6] School of Dentistry, National Yang-Ming University, Taipei 112, Taiwan. [7] Graduate Institute of Medical Science, National Defense Medical Center, Taipei 114, Taiwan. [8] Institute of Molecular and Genomic Medicine, National Health Research Institutes, Zhunan, Miaoli County 350, Taiwan. [9] Core Facility Center, Office of R&D, Taipei Medical University, Taipei 110, Taiwan. Correspondence and requests for materials should be addressed to J.W.S. (email: shihjw@tmu.edu.tw) or H.J.K. (email: hkung@nhri.org.tw)

Hypoxia is a common feature of rapidly growing solid tumours, tightly associated with tumour metastasis and poor prognosis, and a contributor to malignant progression and aggressive phenotype in many cancer types. Hypoxia-inducible factor-1 (HIF-1), a heterodimer consisting of α- and β-subunits, is a key regulator of the cellular response to hypoxia. Under hypoxic conditions, the HIF-1α subunit is stabilized and translocated to the nucleus where it forms a stable HIF-1 complex, specifically bound to the promoter regions of HIF-1 target genes and thereby inducing gene transcription[1]. Proteins encoded by HIF-1 target genes are involved in multiple aspects of tumourigenesis, including glucose and energy metabolism, proliferation, cancer stem-like properties, angiogenesis, invasion and metastasis[1,2]. The activation of HIF-1 pathways is associated with an aggressive tumour phenotype and poor clinical outcome in numerous cancer types, including oral cancer[3]. Oral squamous cell carcinoma (OSCC) represents one of the most common malignancies worldwide with a high mortality rate mainly due to lack of early detection markers, frequent association with metastasis and aggressive phenotype. Thus, there is an urgent need to identify biomarkers and therapeutic targets for this disease.

The recent discovery of long noncoding RNAs (lncRNAs) has gained widespread attention as a new layer of regulation in biological processes. It has been shown that lncRNAs function primarily through their interactions with cellular macromolecules, such as chromatin DNA, proteins and RNAs[4–6]. Accumulating evidence links mutations and dysregulations of lncRNAs to various human diseases ranging from neurodegeneration to cancer[7–10]. To date, numerous cancer-associated lncRNAs are reported to modulate tumour growth, invasion and metastasis, and have been implicated as potential alternative biomarkers and therapeutic targets for cancer[9,10]. Although several hypoxia responsive lncRNAs have been reported to play important roles in HIF-1 pathway regulation and tumorigenesis processes[11–13], none was shown to modulate HIF-1 transactivation ability through direct interaction with it.

MIR31HG (also known as LOC554202; Accession number: NR_027054) is the host gene of microRNA (miRNA), miR-31, which is mapped to the first intron of MIR31HG[14]. Accordingly, the transcriptional regulation of miR-31 was shown to follow that of its host gene MIR31HG and both are epigenetically regulated by promoter methylation[14]. MIR31HG is significantly upregulated in lung[15], breast[16] and pancreatic cancer[17], whereas MIR31HG knockdown inhibited cell growth, invasion[16,17] and induced a p16$^{INK4A}$-dependent senescence phenotype[18]. By contrast, MIR31HG downregulation is reported in colorectal[19] and gastric cancer[20]. The functional role and the underlying mechanism of MIR31HG in hypoxia-associated cancer progression remain largely elusive.

In a comprehensive survey of lncRNAs regulated by hypoxia, we identify MIR31HG as a species that is significantly upregulated. Here we describe a previously unrecognized role of the mature, spliced form of MIR31HG as a co-activator of HIF-1α that activates the pseudohypoxia signature required for hypoxia-induced metabolic reprogramming, sphere-forming ability and metastatic potential. As this RNA species does not contain miR-31 sequence and functions independently, we refer to it as LncHIFCAR (long noncoding HIF-1α co-activating RNA). We also uncover the upregulation of LncHIFCAR/MIR31HG in oral carcinoma and the clinical relevance of LncHIFCAR as an independent adverse prognostic predictor for the cancer progression. Given its significance in the HIF-1 signalling pathway, LncHIFCAR/MIR31HG represents a novel and potential therapeutic target for the treatment of oral carcinoma.

## Results

**The expression of lncRNA LncHIFCAR is induced by hypoxia**. To identify lncRNAs involved in HIF-1 signalling pathway and hypoxia-associated cancer progression, we initially selected 37 cancer-associated lncRNAs (listed in Supplementary Data 1) according to previous reports[8,21] and examined their expression profiles in HeLa cells before and after hypoxia treatment using quantitative real-time PCR (qRT–PCR). Validation of a panel of known hypoxia-inducible protein-coding genes by qRT–PCR confirmed the robustness of our screenings (Supplementary Fig. 1). Compared with non-hypoxic controls, several lncRNAs with a >2-fold alteration in expression were identified under hypoxic conditions (Fig. 1a). Notably, several hypoxia-responsive lncRNAs reported by recent studies, such as H19 (ref. 22) and lncRNA-UCA1 (ref. 23), were also identified in this test. In addition, taking advantage of the hypoxia-mimetic agent cobalt chloride, known to stabilize HIF1α by inactivating prolyl hydroxylases and induce a cellular pseudo-hypoxic state[24], we examined the expression of the lncRNAs and identified five of them upregulated by >2-fold after treatment of cobalt chloride in HeLa cells (Fig. 1b). These lncRNAs include four of those induced in physical hypoxia. LncHIFCAR/MIR31HG was the most substantially upregulated lncRNA in response to cobalt chloride treatment and displayed a time- and dose-dependent induction (Supplementary Fig. 2). Similar to the chemically induced pseudo-hypoxic conditions, a time-dependent increase of LncHIFCAR was also observed in physical hypoxic HeLa cells (Fig. 1c), confirming LncHIFCAR as a specific, hypoxia-inducible lncRNA that is possibly involved in HIF-1α signalling pathway.

**LncHIFCAR is a prognostic biomarker for OSCC**. To evaluate the clinical significance of LncHIFCAR in cancer progression, we first queried the Oncomine database (www.oncomine.com) to systematically assess the relative LncHIFCAR expression in different cancer types (normal versus cancer). Several types of cancer were found to exhibit a significant up-regulation of LncHIFCAR, including OSCC ($P = 2.2 \times 10^{-17}$, Student's t-test; Fig. 1d)[25], colon adenocarcinoma, rectal adenocarcinoma, thyroid cancer[26,27] and breast cancer[28] (Supplementary Fig. 3). Consistent with the published OSCC data set, in a panel of 15 matched pairs of clinical specimens containing OSCC tumours and the surrounding non-cancerous mucosa tissues, LncHIFCAR was substantially upregulated in the tumour samples with a >2-fold overexpression in 8 out of the 15 (53%) samples (Fig. 1e). Notably, as OSCC is the most common type of head and neck SCC (HNSCC), we also surveyed the RNA sequencing data of The Cancer Genome Atlas (TCGA) HNSCC study using cBioPortal platform[29,30] and found a significantly upregulated LncHIFCAR expression in HNSCC tumour samples (Supplementary Fig. 4). To assess the clinical significance of LncHIFCAR in OSCC, we next examined the relationship between LncHIFCAR expression level and the clinicopathological characteristics of 42 OSCC samples. High LncHIFCAR expression was significantly associated with age ($P = 0.037$, Fisher's exact test) and advanced tumour grade ($P = 0.01$, Fisher's exact test), whereas no significant relationship with any other clinicopathological characteristics was observed (Supplementary Table 1). Kaplan–Meier survival analysis was then performed to compare the outcomes of patients dichotomized by LncHIFCAR expression. Patients with high LncHIFCAR expression level had a significantly worse overall survival (OS, $P = 0.021$, log-rank test; Fig. 1f) and recurrence-free survival (RFS, $P = 0.004$, log-rank test; Fig. 1g) than those with low LncHIFCAR expression. Moreover, similar to the prognostic

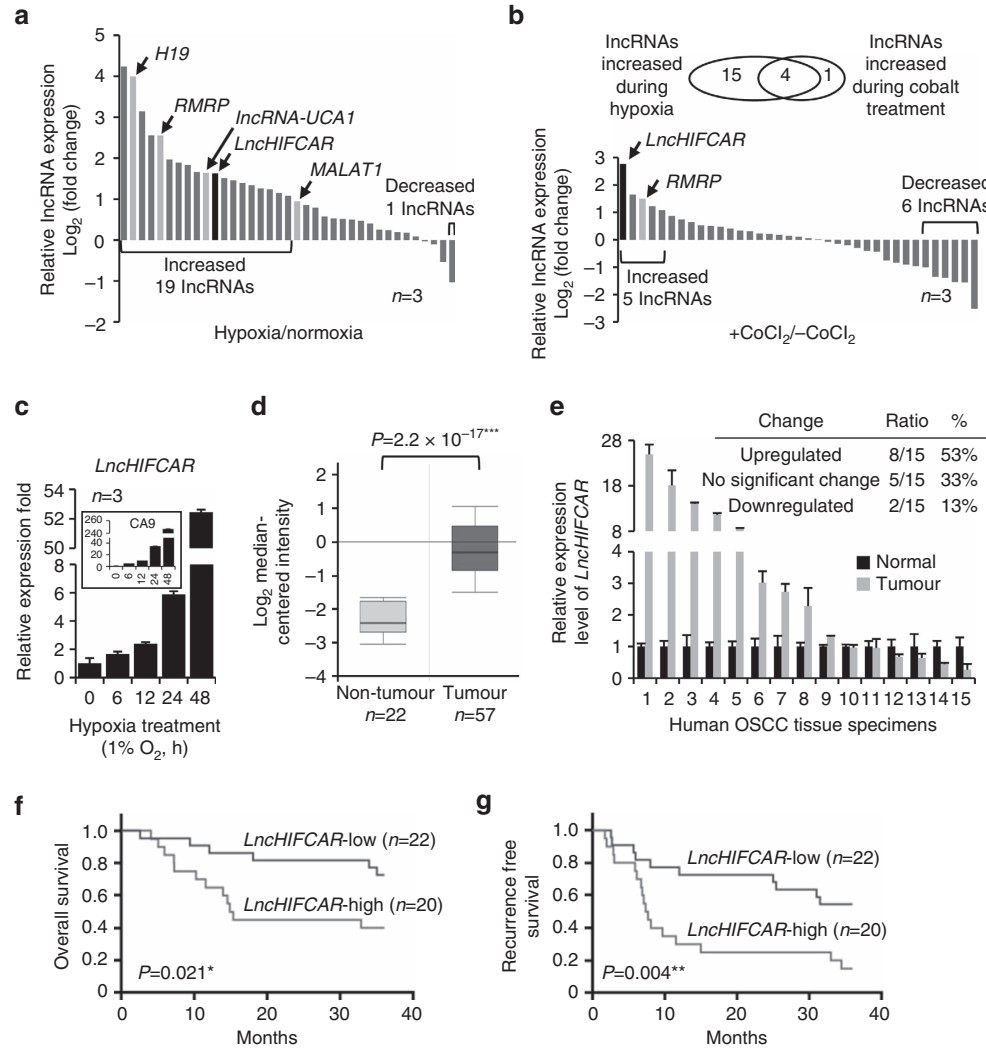

**Figure 1 | *LncHIFCAR* expression is induced upon hypoxia and highly upregulated in human oral carcinoma with prognostic value.** (**a,b**) The expression profiling of 37 cancer-associated lncRNAs in HeLa cells under hypoxia (1% $O_2$ for 16 h; **a**) or treated with hypoxia-mimetic agent cobalt chloride (100 μM for 16 h; **b**) was assessed by quantitative real-time PCR and normalized against 18S rRNA. Each bar represents the relative expression of each lncRNA after treatment. The Venn diagram shows the number of lncRNAs that were upregulated in response to hypoxia or cobalt chloride treatment. (**c**) Physical hypoxia induced the expression of *LncHIFCAR* in a time-dependent manner within 48 h. HeLa cells were incubated in 1% oxygen for the indicated time period. Real-time PCR analyses determining the expression levels of *LncHIFCAR* and the intrinsic hypoxia marker *CA9* mRNA were shown and normalized to 18S rRNA level. (**d**) *LncHIFCAR* expression in OSCC patient samples analysed in Peng Head-Neck data set (GEO: GSE25099) from Oncomine database. The *LncHIFCAR* expression is presented as box-plot diagrams, with the box encompassing 25th–75th percentile. Solid horizontal black line represents the median while error bars indicate the 10th to 90th percentile. Statistical analyses between different patient groups were examined by Student's *t*-test, ***$P < 0.001$. (**e**) *LncHIFCAR* expression in 15 pairs of OSCC tissues (Tumour) and their normal counterpart tissues (Normal) was analysed by quantitative real-time PCR and normalized to *RPLP0*. The *LncHIFCAR* expression showing a more than twofold change between normal and tumour samples was considered significant. (**f,g**) Kaplan–Meier curves of OS (**f**) and RFS (**g**). Patients were grouped into *LncHIFCAR*-Low or *LncHIFCAR*-High based on the *LncHIFCAR* expression value. The cutoff point is set as the value yielding maximum sum of sensitivity and specificity for RFS analysis. The *P*-values was determined by log-rank test, *$P < 0.05$ and **$P < 0.01$. (**c,e**) Graphs show mean ± s.d. *n*, the number of independent experiments.

values of lymph node metastasis (N0 versus N1–3) and tumour differentiation (G1 versus G2 + G3), the univariate cox regression analysis indicated high *LncHIFCAR* expression as another strong prognostic predictor for poor OS (hazard ratio (HR) = 2.701, $P = 0.037$) and RFS (HR = 3.686, $P = 0.002$, Table 1). To further verify the robustness value of *LncHIFCAR* expression, multivariate analysis was performed to determine risk assessment related to OS and RFS. Most remarkably, high expression of *LncHIFCAR* was identified as an independent prognostic factor for RFS (HR = 3.5, $P = 0.012$, multivariate analysis; Table 1). Together, our data revealed that *LncHIFCAR* is overexpressed in OSCC/HNSCC, and that the level of

*LncHIFCAR* could serve as an independent prognostic indicator of clinical outcomes in OSCC patients.

**LncHIFCAR contributes to cancer progression.** To evaluate the possible role of *LncHIFCAR* in oral cancer, we first analysed the level of *LncHIFCAR* in a panel of OSCC cell lines, OECM1, OC-2, HSC-3 and SAS. Compared with OEC-M1 and OC-2 cells established from primary OSCC tumours that exhibit minimal invasion capacity[31], the highly invasive HSC-3 and SAS cells expressed higher *LncHIFCAR* level (Fig. 2a), indicating a possible association of *LncHIFCAR* with the invasion ability of OSCC cell

**Table 1 | Univariate and multivariate Cox regression analyses of OS and RFS in OSCC patients.**

| Covariates | Univariate analysis | | Multivariate analysis | |
|---|---|---|---|---|
| | HR (95% CI) | *P*-value | HR (95% CI) | *P*-value |
| *OS* | | | | |
| Age (≤50 versus >50 years) | 2.299 (0.872–6.062) | 0.092 | 1.860 (0.580–5.965) | 0.296 |
| T status (≤4 cm versus >4 cm) | 2.820 (1.137–6.991) | 0.025* | 2.417 (0.584–9.991) | 0.222 |
| Stage (I + II versus III + IV) | 3.221 (1.156–8.979) | 0.025* | 1.045 (0.164–6.636) | 0.962 |
| Tumour differentiation (G1 versus G2 + G3) | 2.556 (1.034–6.319) | 0.042* | 1.547 (0.517–4.628) | 0.434 |
| Lymph node metastasis (N0 versus N1–3) | 3.222 (1.259–8.239) | 0.014* | 3.193 (0.845–12.058) | 0.086 |
| *LncHIFCAR* expression level (high versus low) | 2.701 (1.058–6.896) | 0.037* | 2.239 (0.719–6.966) | 0.163 |
| *RFS* | | | | |
| Age (≤50 versus >50 years) | 1.575 (0.713–3.479) | 0.261 | 0.967 (0.315–2.967) | 0.954 |
| T status (≤4 cm versus >4 cm) | 1.569 (0.697–3.526) | 0.276 | 1.827 (0.457–7.299) | 0.394 |
| Stage (I + II versus III + IV) | 1.986 (0.897–4.394) | 0.090 | 0.602 (0.083–4.339) | 0.615 |
| Tumour differentiation (G1 versus G2 + G3) | 2.709 (1.242–5.909) | 0.012* | 1.453 (0.608–3.469) | 0.401 |
| Lymph node metastasis (N0 versus N1–3) | 3.043 (1.374–6.735) | 0.006** | 4.147 (0.869–19.770) | 0.074 |
| *LncHIFCAR* expression level (high versus low) | 3.686 (1.616–8.409) | 0.002** | 3.500 (1.317–9.302) | 0.012* |

CI, confidence interval; G1, well-differentiated; G2, moderately differentiated; G3, poorly differentiated; HR, hazard ratio; OS, overall survival; OSCC, oral squamous cell carcinoma; RFS, recurrence-free survival.
Statistically significant (*P<0.05 and **P<0.01).

lines. Similar to HeLa cell, *LncHIFCAR* expression is also substantially induced in SAS cells upon chemical-induced pseudohypoxia or physical hypoxia in a dose- and time-dependent manner (Supplementary Fig. 5).

Cellular response to hypoxia is implicated in many critical aspects of cancer progression, including invasion, metastasis, stem properties maintenance and metabolism reprogramming. To further characterize the biological significance of *LncHIFCAR* in oral tumorigenesis, we knocked down this lncRNA with two independent small interfering RNAs (siRNAs). In addition, we used short hairpin RNA (shRNA)-expressing plasmids to generate stable *LncHIFCAR* knockdown clones (sh-*HIFCAR*) and vector controls (sh-Ctrl) in SAS cells, and examined the hypoxia-associated phenotypes in these cells. Notably, although miR-31 is located within the first intron of *LncHIFCAR* and shares the same transcription promoter[14], the *LncHIFCAR* knockdown shRNA/siRNA was designed to target the exon region and thus only specifically downregulated *LncHIFCAR* but not miR-31, as determined by quantitative PCR (Fig. 2b and Supplementary Fig. 6a). As depicted in Fig. 2c and Supplementary Fig. 6b, the transwell assays showed that *LncHIFCAR* knockdown resulted in a significantly impaired migration and invasion ability of SAS cells under hypoxia, echoing the positive correlation between *LncHIFCAR* levels and the invasion ability of different OSCC cell lines (Fig. 2a). Consistent with previous report that knockdown of *LncHIFCAR* impeded the growth of breast cancer cell[16], the *LncHIFCAR*-knockdown oral cancer cells also grew more slowly than the vector control (Fig. 2d and Supplementary Fig. 6c), whereas *LncHIFCAR* overexpression promoted the cell growth of SAS and 293T cells (Supplementary Fig. 6d) under normoxic and hypoxic conditions. Notably, the differences in cell proliferation became markedly more profound under hypoxic conditions (2.1-fold, Fig. 2d, right panel) than under normoxic conditions (1.2-fold, Fig. 2d, left panel), revealing a crucial role of *LncHIFCAR* in hypoxic cell growth. In addition, with normalization to cell number, we detected a significant reduction of hypoxia-induced glucose uptake (Fig. 2e and Supplementary Fig. 6e) and lactate production (Fig. 2f and Supplementary Fig. 6f) in the *LncHIFCAR*-knockdown SAS cells, suggesting that *LncHIFCAR* is essential for glycolysis in hypoxia and confers a proliferation advantage to hypoxic oral cancer cells. We then tested anchorage-independent sphere formation and found that, compared to the control cells, *LncHIFCAR*

knockdown cells exhibited significantly weaker tumour sphere-forming ability (Fig. 2g and Supplementary Fig. 6g). Together, these data suggest that *LncHIFCAR* acts as a critical mediator in hypoxia-associated tumorigenesis including migration, invasion, hypoxic cell growth, metabolic regulation and the sphere-forming ability in OSCC cells.

**LncHIFCAR activates HIF-1 transcriptional network**. HIF-1, consisting of HIF-1α and HIF-1β subunits, is the primary factor driving cellular response to hypoxia by activating the expression of target genes involved in critical steps of phenotype changes in cancer progression, including angiogenesis (*VEGF*, vascular endothelial growth factor), mitochondrial function (*BNIP3*, BCL2/Adenovirus E1B 19 kDa interacting protein 3), metabolism reprogramming (*GLUT1/SLC2A1*, glucose transporter 1; *LDHA*, lactate dehydrogenase A; *CA9*, carbonic anhydrase 9; *PDK1*,pyruvate dehydrogenase kinase isozyme 1), invasion (*LICAM*, L1 cell adhesion molecule) and metastasis (*LOXL2*, lysyl oxidase homologue 2)[32,33]. Given its role in the hypoxia-associated cancer phenotypes, we next investigated the impact of *LncHIFCAR* on the expression of HIF-1 target genes. Strikingly, overexpression of *LncHIFCAR* but not *RMRP*, another hypoxia-inducible lncRNA, induced the activation of HIF-1 target genes, known as pseudohypoxia, without a significant change of *HIF1A* messenger RNA in HeLa cells under normoxic condition (Fig. 3a). Furthermore, induction of the pseudohypoxia signature showed a positive correlation with the levels of *LncHIFCAR* (Fig. 3b) and appeared to be HIF-1α specific, as the expression of non-HIF-1 target *RPL13A* (ribosomal protein L13a) and HIF-2 target *CITED2* (CBP/p300-interacting transactivator 2) remained unchanged upon *LncHIFCAR* overexpression (Fig. 3b). Reciprocally, knockdown of *LncHIFCAR* significantly attenuated the activation of HIF-1-specific target genes in chemical-induced pseudo-hypoxic HeLa cells (Fig. 3c and Supplementary Fig. 7a), as well as in the SAS cells under physical hypoxic condition (Fig. 3d and Supplementary Fig. 7b), indicating that *LncHIFCAR* plays a central role in the activation of HIF-1 target genes.

Remarkably, transcriptional profiling of the tumour spheres suggested that the *LncHIFCAR*-mediated HIF-1 activation may functionally contribute to the sphere-forming ability in OSCC cells. When we examined the expression of *LncHIFCAR* and

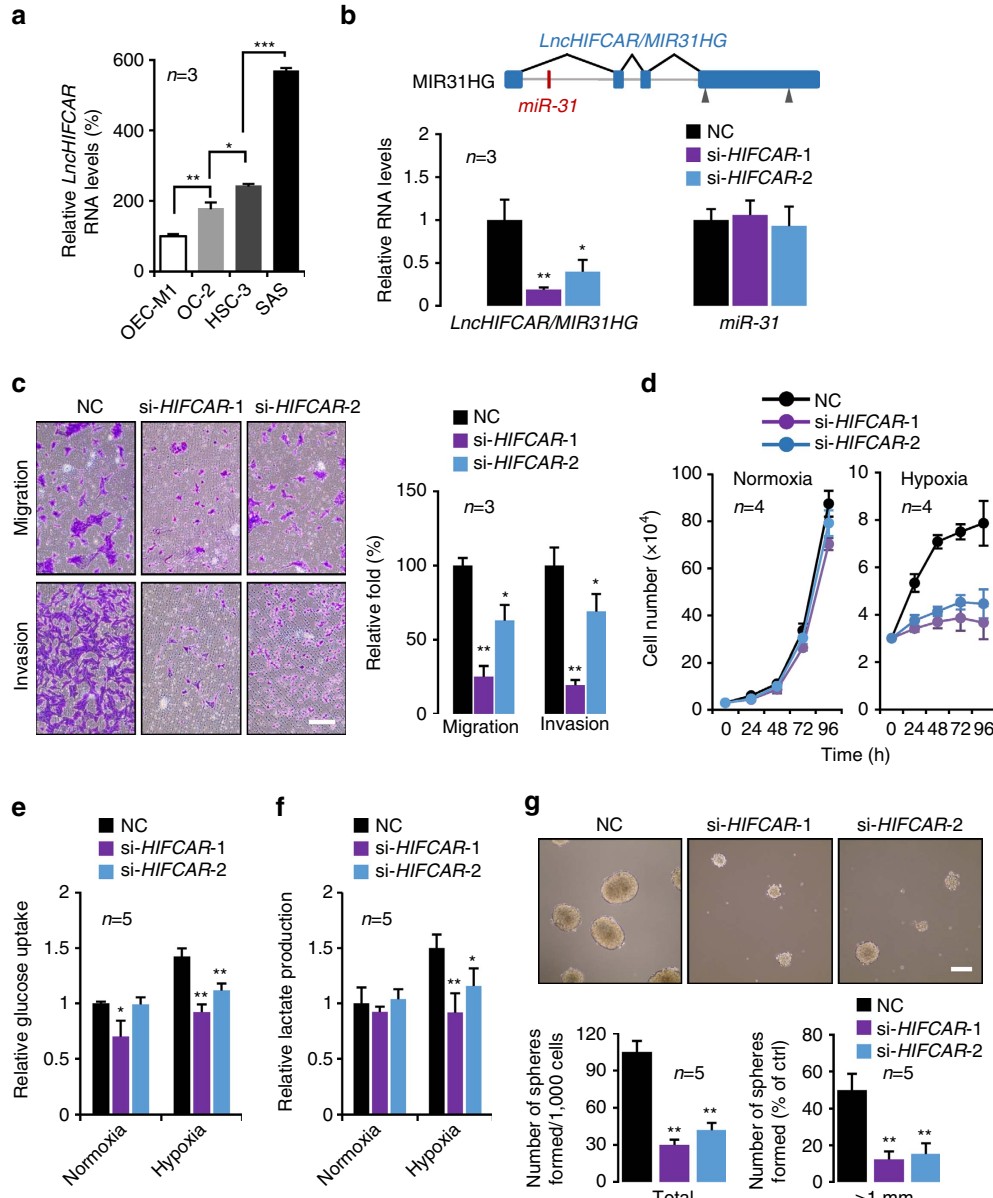

**Figure 2 | *LncHIFCAR* functions as an oncogene driving oral cancer progression.** (**a**) Expression levels of *LncHIFCAR* in oral cancer cell lines. The levels of *LncHIFCAR* were normalized against *RPLP0* mRNA level and are presented relative to the expression in OEC-M1. (**b**) Establishment of *LncHIFCAR*-knockdown SAS clones. Upper panel: schematic representation of the genomic structure of *LncHIFCAR/MIR31HG*. The targeting sites of the designed siRNAs are indicated by gray arrowhead. *LncHIFCAR/MIR31HG* is composed of four exons (filled blue boxes) and the miR-31, located within intron 1, is shown with a red horizontal bar. Lower panel: *LncHIFCAR* and miR-31 expression levels in siRNA-transfected SAS cells were analysed by qPCR and normalized to *GAPDH* and *U6* level, respectively. (**c**) *LncHIFCAR* promotes oral cancer cell invasion and migration. Transwell invasion and migration assays of siRNA-transfected SAS cells were performed. Representative photos and quantitative analysis are shown. Scale bar, 100 µm. (**d**) Cell growth curve of the siRNA-transfected SAS cells under normoxia or hypoxia. (**e,f**) Knockdown of *LncHIFCAR* reduces hypoxia-induced glucose uptake (**e**) and lactate production (**f**). The glucose and lactate levels in the culture media of the siRNA-transfected SAS cells were measured after 16 h of normoxia or hypoxia treatment. The data are presented as fold difference compared with the level of negative control siRNA-transfected cells in normoxia. (**g**) Knockdown of *LncHIFCAR* reduces the sphere-forming ability of SAS cells. The siRNA-transfected SAS cells were grown in suspension culture to form sphere. Representative phase-contrast microscopic images of the cell aggregates are displayed (scale bar, 300 µm). Bar graph represents the total number of spheres or the relative number of spheres with diameter >1 mm. Graphs show mean ± s.d. NC, negative control. *n*, the number of independent experiments performed; Student's *t*-test, *$P < 0.05$, **$P < 0.01$ and ***$P < 0.001$.

the HIF-1 target genes in HSC-3 and SAS parental and their sphere-forming cells, profound upregulation of *LncHIFCAR* levels accompanied with HIF-1 target gene induction was detected in the sphere-forming cells (Fig. 3e). Our findings of *LncHIFCAR*'s participation in the anchor-independent sphere formation (Fig. 2g and Supplementary Fig. 6g), combined with

*LncHIFCAR*-dependent HIF-1 target activation (Fig. 3c,d and Supplementary Fig. 7a,b) suggest *LncHIFCAR* may facilitate cellular adaption to the developing hypoxia at the core of the spheres. Using the immmunohistochemical hypoxia marker pimonidazole, we observed a hypoxic gradient in the spheres (Fig. 3f and Supplementary Fig. 8). Florescence staining with

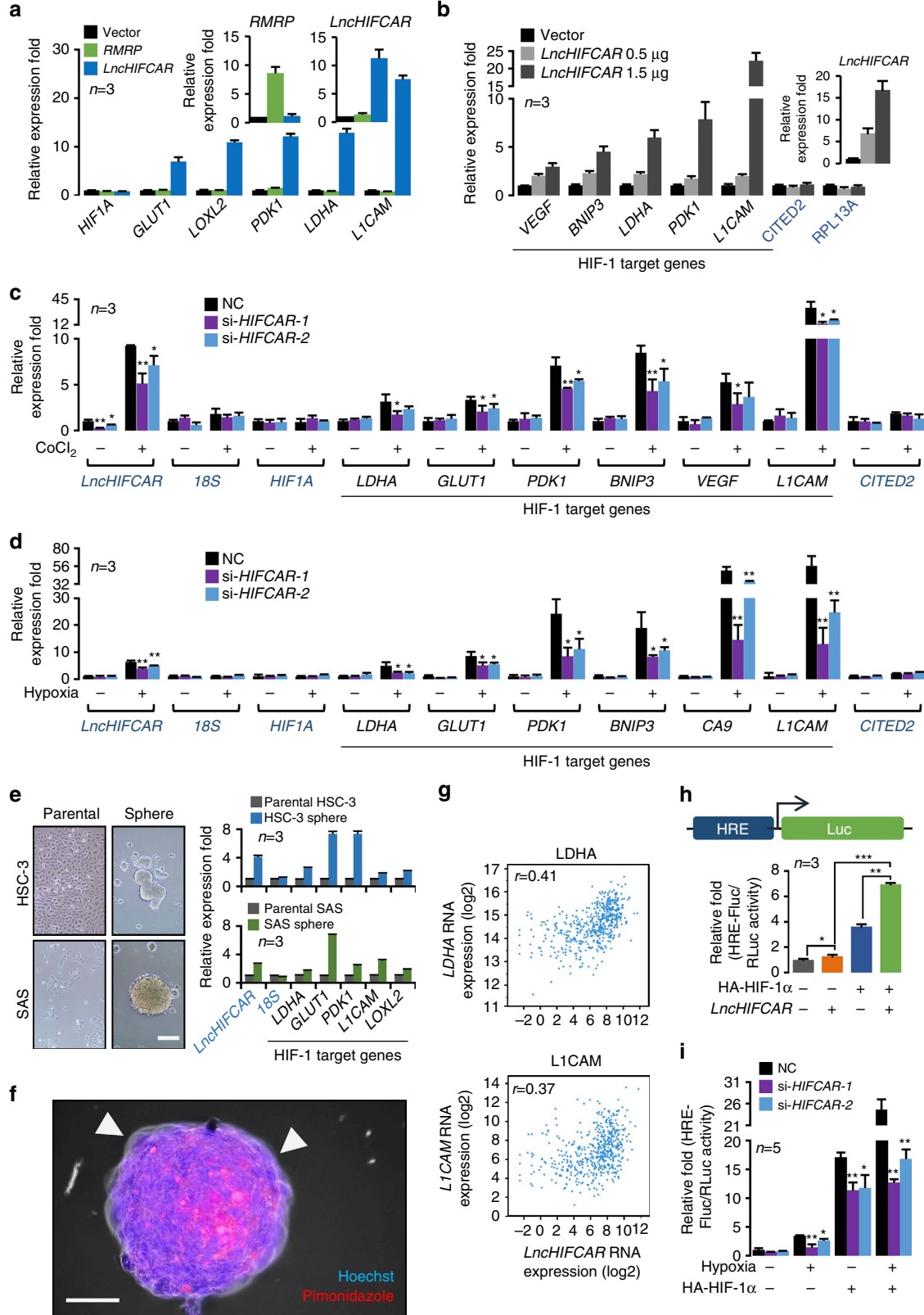

propidium iodide (PI) identifies necrotic cells in these hypoxic core region of spheres (Supplementary Fig. 9). PI counterstaining of spheres with comparable size showed that knockdown of *LncHIFCAR* induced more necrotic cell death (Supplementary Fig. 10), suggesting that *LncHIFCAR* knockdown cells were

unable to adapt to the developing hypoxia at the core of the spheres. Collectively, these results underscore a crucial role of *LncHIFCAR* in the activation of HIF-1 target genes, hypoxia adaption and sphere formation that may further contribute to tumour development.

To further validate the clinical relevance of *LncHIFCAR* with HIF-1 targets expression, we queried RNA sequencing[29,30] and mRNA microarray data[25,28] in published data sets from a variety of cancer studies. Congruent with our findings, HIF-1 target genes *LDHA* (Pearson's correlation $r = 0.41$) and *L1CAM* ($r = 0.37$) showed a positive correlation with the expression of *LncHIFCAR* in the TCGA HNSCC provisional cohort[29,30] ($n = 522$; Fig. 3g), whereas no significant correlation with the *HIF1A* mRNA levels were observed ($r = 0.2$). Similarly, the mRNA of other HIF-1 targets such as *LOXL2*, *PGK1*, *VEGFA* and *GLUT1* were also significantly elevated in human OSCC[25], invasive ductal breast carcinoma[28], colon and rectal adenocarcinoma that highly express *LncHIFCAR*, but not *HIF1A* mRNA (Table 2). On the other hand, the expression of HIF-2 target *CITED2* was not significantly altered in the above-cancer tissues (Table 2), revealing a specific correlation of *LncHIFCAR* with the HIF-1 transcriptional network.

We next determined whether *LncHIFCAR* functions through a direct effect on HIF-1 transactivation potency. A HIF-1α reporter plasmid (HRE-FLuc) containing three hypoxia-response elements (HREs) and firefly luciferase coding sequences was used for the promoter-activity assay. Ectopic expression of HIF-1α (3.62-fold) or *LncHIFCAR* (1.26-fold) alone increased HIF-1 transcriptional activity, whereas co-expression of HIF-1α and *LncHIFCAR* synergistically enhanced the promoter activity (6.94-fold; Fig. 3h). In a reciprocal experiment, *LncHIFCAR*-silencing led to a significant reduction in hypoxia-stimulated or ectopic HIF-1α-induced HIF-1 transcriptional activation (Fig. 3i and Supplementary Fig. 11). Taken together, these data suggest that *LncHIFCAR* augments the transcriptional activity of HIF-1α and serves as a critical regulator of HIF-1 transcriptional network, which enhances the anchorage-independent growth and provides a growth advantage for tumour cells for hypoxia adaptation.

***LncHIFCAR* acts as a HIF-1α co-activator**. The major mechanism of HIF-1 activation in hypoxia is attributed to the stabilization of HIF-1α protein. We found that neither over-expression nor knockdown of *LncHIFCAR* had a significant impact on the protein level of HIF-1α either under normoxic, chemical-induced pseudo-hypoxic or physical hypoxic conditions (Fig. 4a and Supplementary Fig. 12). In addition, western blotting analysis of nuclear and cytosolic fractions revealed no detectable changes in the hypoxia-induced HIF-1α nuclear accumulation in *LncHIFCAR*-knockdown cells compared with control cells (Supplementary Fig. 13). Meanwhile, there is no noticeable difference in the subcellular localization of the *LncHIFCAR* lncRNA in normoxic and hypoxic conditions (Supplementary

Fig. 14). These results suggest a mechanism for *LncHIFCAR*-mediated HIF-1 activation independent of protein stabilization and translocation of HIF-1α.

Under normoxia conditions, the expression of HIF-1α proteins is generally low. We asked whether this low HIF-1α expression is required for *LncHIFCAR*-induced pseudohypoxia signature and HIF-1 transactivation as seen in Fig. 3a,b,h. We found that they both were reverted by HIF-1α knockdown (Supplementary Fig. 15), indicating the critical role of HIF-1α in *LncHIFCAR*-mediated HIF-1 target gene regulation. Therefore, we next examined the physical interaction of *LncHIFCAR* and HIF-1α. Using nuclear extracts of SAS cells, RNA immunoprecipitation assay showed a robust and specific enrichment of *LncHIFCAR* co-precipitated within the HIF-1α immunocomplex in cells under hypoxia, whereas no enrichment was detected for other RNA molecules such as *GAPDH* mRNA, 18S rRNA, a comparable sized lncRNA *PCAT1*, and hypoxia-inducible *RMRP* and *MALAT1* lncRNAs (Fig. 4b). The physical interaction between *LncHIFCAR* and HIF-1α was confirmed by RNA pull-down assay using hypoxic cell nuclear extracts incubating with *in vitro* transcribed biotinylated *LncHIFCAR* (Fig. 4c). With purified recombinant HIF-1α protein in the *in vitro* binding assay, we further validated that *LncHIFCAR* associated with HIF-1α through direct binding (Fig. 4d).

To delineate the structural determinants for the association between *LncHIFCAR* and HIF-1α, RNA pull-down assays were performed with a series of *LncHIFCAR* truncated fragments. Both the 5′-terminal (nucleotides 1–500) and 3′-terminal (nucleotides 1,500–2,166) regions were found to be associated with HIF-1α (Fig. 4d). As expected, the HIF-1α association was abolished when using a truncated *LncHIFCAR* (nucleotides 501–1,500) that lacked the two binding regions (Fig. 4e), confirming that the critical HIF-1α-interaction regions in *LncHIFCAR* reside at the 5′-terminal (nucleotides 1–500) and 3′-terminal (nucleotides 1,500–2,166) sections. To further explore functional relevance of the *LncHIFCAR*/HIF-1α interaction, the full-length and deletion mutant *LncHIFCAR* (501–1,500) were overexpressed in SAS cells. As shown in Fig. 4f and Supplementary Fig. 16, relative to full-length *LncHIFCAR*, the ability of the HIF-1α binding-deficient mutant *LncHIFCAR* (501–1,500) to induce HIF-1 target genes and HIF-1 responsive luciferase reporter was severely impaired. This result suggests that the HIF-1α-binding regions functionally contribute to the *LncHIFCAR*-mediated HIF-1 target induction. We next mapped HIF-1α domain required for *LncHIFCAR* binding by generating HIF-1α bHLH (basic helix–loop–helix; amino acids 1–80), PAS-A (Per-ARNT-Sim-A; amino acids 81–200), PAS-B (Per-ARNT-Sim-B; amino

**Figure 3 | *LncHIFCAR* is required for the hypoxia-induced activation of HIF-1 target genes and the HIF-1-dependent transactivation.** (**a**) The RNA levels of *HIF1A* and HIF-1 target genes in HeLa cells transfected with vector, *LncHIFCAR*- or RMRP-expressing plasmids for 48 h measured by qRT–PCR. (**b**) The RNA levels of HIF-1 target genes in HeLa cells transfected with different amount of the *LncHIFCAR*-expressing plasmid measured by qRT–PCR. (**c**) HIF-1 target genes expression in HeLa cells transfected with the indicated siRNAs. Twenty-four hours after transfection, the cells were treated by $CoCl_2$ for 16 h, followed by qRT–PCR analysis with normalization against *RPLP0* level. (**d**) qRT–PCR analysis of HIF-1 target genes expression (normalized to *RPLP0*) in siRNA-transfected SAS cells under 24 h treatment of normoxia or hypoxia. (**e**) The expression of *LncHIFCAR* and HIF-1 target genes in tumour spheres derived from HSC3 and SAS cells (normalized to *GAPDH*), compared with monolayer cultures of the parental cells. Insets, representative phase contrast microscopic images of the monolayer and sphere cells (scale bar, 200 μm). The 18S rRNA level is served as a control. (**f**) Developing hypoxic region at the core of the spheres derived from SAS oral cancer cells was observed in immunostaining preparations using an exogenous hypoxia marker pimonidazole (scale bar, 100 μm). (**g**) The expressional correlation between *LncHIFCAR* and HIF-1 target genes (*LDHA* and *L1CAM*) in the HNSCC (TCGA, Provisional) study ($n = 522$) were surveyed using cBioPortal platform. The corresponding correlation plots are shown with Pearson's coefficients. (**h,i**) HIF-1α-responsive luciferase reporter assays in *LncHIFCAR*-overexpressing (**h**) or *LncHIFCAR*-knockdown (**i**) cells presented as relative value with normalization against *Renilla*-Luc activity. (**h**) A plasmid containing HIF-1α-responsive luciferase reporter was co-transfected with empty vector, HIF-1α or *LncHIFCAR* into HeLa cells for the reporter assay. (**i**) Reporter assay performed in siRNA-transfected HeLa cells co-transfected with the indicated plasmids for 24 h, followed by 24 h treatment of hypoxia or normoxia. Graphs show mean ± s.d. NC, negative control. *n*, the number of independent experiments performed; Student's *t*-test, *$P < 0.05$, **$P < 0.01$ and ***$P < 0.001$.

**Table 2 | Expression levels of *LncHIFCAR* and HIF-1α target genes in normal versus cancer obtained from the Oncomine microarray data sets.**

| RNA | Expression fold change | | | |
|---|---|---|---|---|
| | Peng head–neck oral cavity SCC (57) versus normal (22) | Zhao breast invasive ductal breast carcinoma (40) versus normal (3) | TCGA colorectal colon adenocarcinoma (101) versus normal (19) | TCGA colorectal rectal adenocarcinoma (60) versus normal (19) |
| *LncHIFCAR* | 4.484*** | 1.693** | 1.916*** | 1.762*** |
| LOXL2 | 1.977*** | 1.866*** | 2.437*** | 2.33*** |
| PGK1 | 1.362*** | 1.889*** | 1.982*** | 2.216*** |
| VEGFA | 1.164* | 2.598*** | 1.054*** | 1.095*** |
| GLUT1 | 1.135* | 2.8*** | 2.017*** | 2.282*** |
| LDHA | 1.414*** | 1.467** | 1.176* | NS |
| HIF1A | 1.188* | NS | NS | NS |
| CITED2 | NS | NS | NS | NS |

HIF-1α, hypoxia-inducible factor-1α; NS, no statistically significant difference (*t*-test, *P* > 0.05); SCC, squamous cell carcinoma.
Statistically significant (*t*-test, *P* < 0.05, **P < 0.01 and ***P < 0.001).

acids 201–329) and TAD (transactivation domain; amino acids 191–445) deletion mutants (Fig. 4g, bottom panel). *In vitro* pull-down assays revealed that *LncHIFCAR* strongly bound to the PAS-B domain of HIF-1α (Fig. 4g and Supplementary Fig. 17). As this domain is responsible for functional dimerization, we then conducted immunoprecipitation of HIF-1α with hypoxic cell lysate to investigate the effect of *LncHIFCAR* on HIF1 complex formation. Knockdown of *LncHIFCAR* reduced the interaction between HIF-1α and both HIF-1β and p300 (Fig. 4h), indicating that *LncHIFCAR* could facilitate the recruitment of HIF-1 complex. Collectively, the data presented in Figs 3 and 4 indicates that *LncHIFCAR* functions as a transcriptional co-activator of HIF-1α. Given that the HIF-1α level in many tumour cells can remain elevated under normoxic conditions[34,35], *LncHIFCAR* may induce pseudohypoxia signature by promoting HIF-1 complex formation under normoxia.

**LncHIFCAR enhances HIF-1 complex binding to the target loci**. To verify whether *LncHIFCAR* directly acts on the target chromatins, we analysed the loci of *LncHIFCAR*-dependent HIF-1 target genes as identified in Fig. 3d by employing chromatin isolation by RNA purification (ChIRP). A set of probes complementary to *LncHIFCAR* was used to pull down the endogenous *LncHIFCAR* from normoxic or hypoxic SAS cells and the promoter regions of known HIF-1α-binding sites were amplified and quantified by quantitative PCR (qPCR). In concordance with the notion that *LncHIFCAR* functions as a HIF-1α co-activator, the ChIRP analysis showed that *LncHIFCAR* is recruited to a subset of the HIF-1 target promoters in hypoxic cells, suggesting overlapping chromatin occupancy of *LncHIFCAR* and HIF-1α under hypoxia (Fig. 5a). As hypoxia-induced target activation is mediated by HIF-1 transcriptional complex composed of HIF-1 heterodimer and the transcription cofactor p300/CBP[2,36], we next examined whether the chromatin binding of *LncHIFCAR* is involved in the recruitment of HIF-1α and the cofactor upon hypoxia. Significantly, the chromatin immunoprecipitation (ChIP) assays using anti-HIF-1α and anti-p300 antibodies showed that these transcriptional factors were enriched on the target promoters in hypoxic cells, whereas knockdown of *LncHIFCAR* significantly reduced this chromatin enrichment (Fig. 5b,c). These results suggest that under hypoxia, *LncHIFCAR* forms a complex with HIF-1α and enhances the chromatin recruitment of HIF-1α and p300 cofactor, thereby promoting the transcriptional reprogramming essential in multiple oncogenic pathways.

**LncHIFCAR promotes metastatic cascade**. With our evidence of *LncHIFCAR* in HIF-1 activation and hypoxia-associated cancer phenotypes, the biological significance of *LncHIFCAR* in oral cancer progression *in vivo* was examined using mouse xenograft model. The vector control and *LncHIFCAR* knockdown SAS cells were genetically modified to express firefly luciferase, and subsequently injected intravenously into nude mice (Day 0). Following tail vein inoculation, the bioluminescence intensity of the lung region was measured weekly up to 6 weeks. Although the vector control cells exhibited strong colonization to the lung with a timely progression, *LncHIFCAR*-knockdown cells showed a drastically reduced ability to colonize in lung (Fig. 6a–c and Supplementary Fig. 18). Verified by histology, a significantly increased incidence of pulmonary metastasis was observed in the mice bearing vector control cells, whereas only rare tumour foci were found in the lungs of mice carrying *LncHIFCAR* knockdown cells (Fig. 6d–f). Notably, the difference in tumour foci size and occurrence cannot be fully accounted for by cell proliferation rate caused by *LncHIFCAR* modulation under normoxia. As judged from the time-lapsed image of Supplementary Fig. 18, control vector-expressing cells colonized to the lung 14 days after tail vein injection. By contrast, the *LncHIFCAR* knockdown xenografts showed significant less lung colonization at the same time point when no apparent change of the injected cell density was observed compared with the control group (Day 14, Fig. 6a,b and Supplementary Fig. 18). Given the critical role of hypoxic signalling in the formation of a premetastatic niche essential for tumour colonization at the distant site[1,2], we propose that *LncHIFCAR*-mediated difference in colonization ability is a combined effect of enhanced HIF-1 signalling activation and increased hypoxic tumour cell growth advantage conferred by this lncRNA. Altogether, these results suggest that *LncHIFCAR* enhanced the metastatic spread of SAS oral cancer cells *in vivo*, further substantiating its oncogenic potential in OSCC progression.

**Discussion**
Hypoxia exists in majority of solid tumour and contributes to local and systemic cancer progression, therapeutic response and poor outcome. Given the pivotal roles of lncRNA in gene expression, the critical involvements of lncRNA in hypoxia-associated malignant progression have been recognized and vigorously studied. Recent reports have identified several hypoxia-responsive lncRNAs. Among them, *lncRNA-PRINS* (ref. 37), *lncRNA-EFNA3* (ref. 38), *lncRNA-UCA1* (ref. 23), *lncRNA-AK058003* (ref. 39), *lncRNA-AK123072* (ref. 40), *lncRNA-NUTF2P3-001* (ref. 41), *linc-p21* (ref. 42),

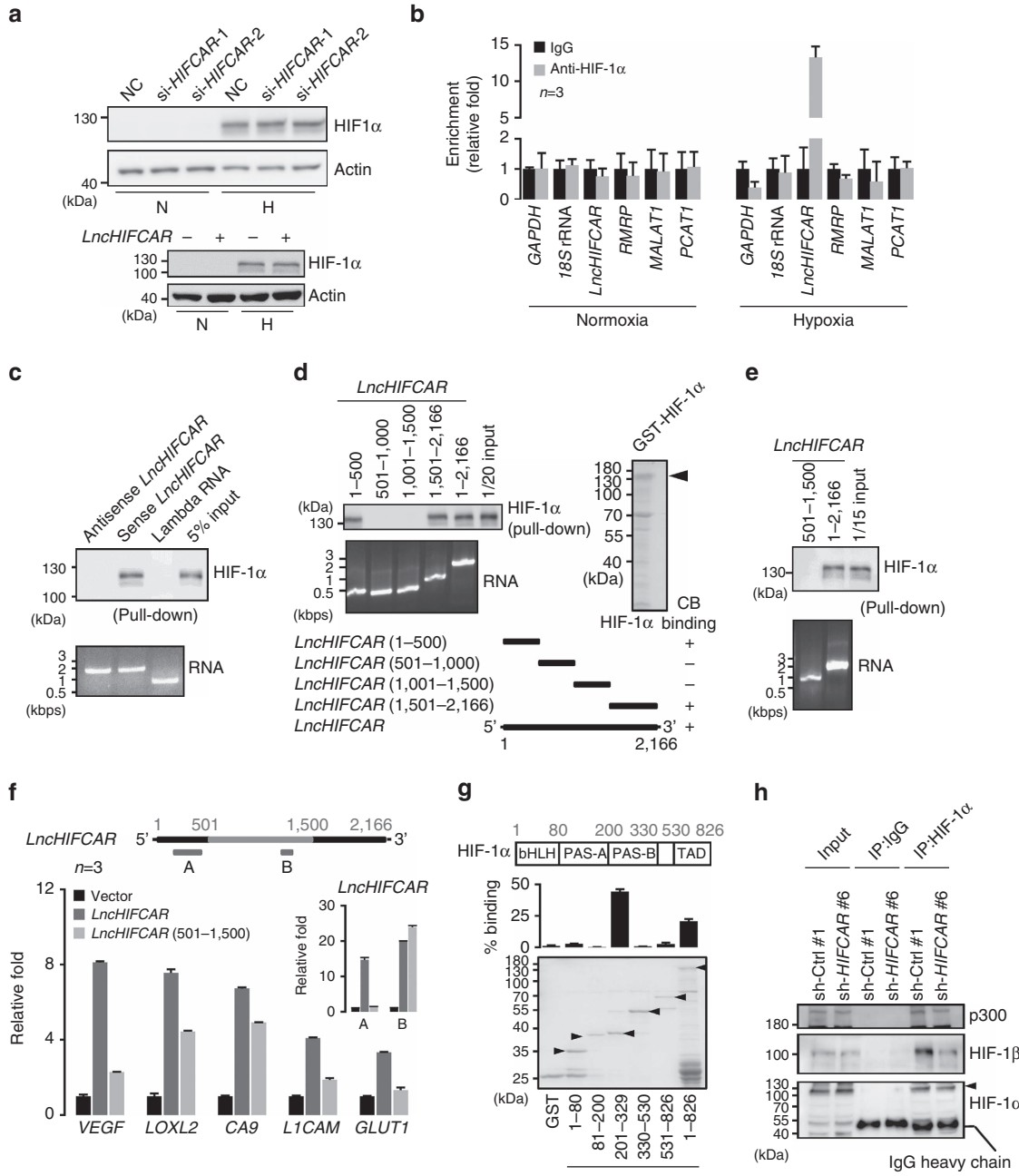

**Figure 4 | lncRNA *LncHIFCAR* associates with HIF-1α and functions as a HIF-1α co-activator.** (**a**) Representative (*n* = 3) western blot analysis of HIF-1α protein levels in *LncHIFCAR*-overexpressing HeLa cells and controls (lower panel), or in siRNA-transfected SAS cells (upper panel) under 24 h of normoxia (N) or hypoxia (H) treatment. NC, negative control. (**b**) lncRNA *LncHIFCAR* enriched in HIF-1α immunoprecipitates under hypoxia. Sixteen hours normoxia- or hypoxia-treated SAS nuclear extracts were immunoprecipated using mouse IgG or anti-HIF-1α antibody. Immunoprecipitation of HIF-1α-associated RNA was validated by qRT–PCR and shown as the relative fold of the RNA enrichment. (**c**) Representative (*n* = 3) immunoblot detection of HIF-1α retrieved by biotinylated *LncHIFCAR* RNA pull-down assay. Biotin-labelled lambda RNA, sense or antisense *LncHIFCAR* RNAs were incubated with 16 h hypoxia-treated HeLa nuclear extracts and then pulled down by streptavidin beads, followed by western blot analysis. (**d,e**) HIF-1α binding domain on *LncHIFCAR*. RNAs corresponding to indicated *LncHIFCAR* fragments were biotinylated and incubated with recombinant HIF-1α protein, followed by streptavidin pull-down as described above. Representative (*n* = 3) immunoblot detection of the associated HIF-1α protein was shown. CB, Coomassie brilliant blue staining. (**f**) HIF-1 target gene expression in SAS cells overexpressing control vector, wild-type or mutant (500–1,500) *LncHIFCAR* RNA. Specific primers are designed for the quantification of wild-type (region A) or mutant (500–1,500) (region B) *LncHIFCAR* levels as shown in the schematic diagram and the primer sequences are provided in Supplementary Data 2. (**g**) *LncHIFCAR* binding domain within HIF-1α. Schematic representation of HIF-1α functional domains and GST-HIF-1α variants are shown at the top. The Coomassie Blue staining showed loading of the proteins and arrowheads mark the GST-HIF-1α truncates. *LncHIFCAR* RNA were pulled down by GST-fusion proteins pre-bound on glutathione-Sepharose beads, followed by qRT–PCR detection of *LncHIFCAR* RNA retrieved as presented as percentage relative to input RNA. bHLH, basic helix–loop–helix; PAS, Per-ARNT-Sim; TAD, transactivation domain. (**h**) *LncHIFCAR* facilitates HIF-1α complex formation. HIF-1α was immunoprecipitated from hypoxic vector control or *LncHIFCAR* knockdown SAS cell extracts. Co-immunoprecipitation of p300 and HIF-1β was reduced in *LncHIFCAR* knockdown cells. Graphs show mean ± s.d. *n*, the number of independent experiments performed.

*MALAT1* (ref. 43), *HOTAIR* (ref. 44), *NEAT1* (ref. 43), *linc-ROR* (ref. 45), *HINCUTs* (ref. 46), *H19* (ref. 22), *WT1* (ref. 47) and *HIF1A-AS2* (*aHIF*) (ref. 48) are upregulated, whereas *lncRNA-LET* (ref. 49) and *ENST00000480739* (ref. 50) are downregulated in response to hypoxia. Interestingly, *lncRNA-SARCC* could differentially respond to hypoxia in a VHL-dependent manner[51]. Most of these hypoxia-regulated lncRNAs are functionally characterized and have profound impact on tumorigenesis in a spectrum of cancer types. However, only few have been fully characterized with respect to the detailed mechanisms of action, with the roles of the majority others remain largely unknown. At epigenetic level, *WT1* modulates histone modification[47], whereas *AK058003* regulates DNA methylation[39]. In addition, lncRNAs may regulate HIF-1 signalling by acting as a miRNA sponge (*lncRNA-NUTF2P3-001* versus miR-3923 (ref. 41) and *linc-ROR* versus miR-145 (ref. 45)) or by functioning as antisense lncRNAs (*aHIF1α*[48] and *HIF2PUT*[52]). Moreover, through lncRNA–protein interaction, hypoxia-responsive lncRNAs, such as *lncRNA-LET*[49], *lincRNA-p21* (ref. 42) and *LINK-A*[53], modulate HIF-1 signalling via their effects on the expression, degradation and phosphorylation of HIF-1α, respectively. Our present study identified *LncHIFCAR* as another hypoxia-inducible lncRNA and described a distinct mechanism, whereby *LncHIFCAR* modulates HIF-1 transcriptional activation independent of the stabilization or cellular translocation of HIF-1α. We found that *LncHIFCAR* is highly elevated in oral cancer, and that the *LncHIFCAR*-dependent activation of HIF-1 modulates target genes expression and thereby confers metabolic reprogramming and a hypoxic proliferation advantage required in several tumour progression steps including sphere-forming ability and metastatic colonization capability (Fig. 6g).

Mechanistically, we showed that *LncHIFCAR* directly interacts with HIF-1α and associates with the chromatin loci of HIF-1 target promoters. Such chromatin complex formation enhanced the recruitment of co-activator p300 and via which facilitated the activation of HIF-1 transcriptional network. This evidence along with the finding that co-expressing *LncHIFCAR* and HIF-1α synergistically enhanced the HIF-1 reporter activity, strongly suggests that *LncHIFCAR* functions as a co-activator of HIF-1α. Of note, *LncHIFCAR/MIR31HG* is also reported to associate with the promoter of its own gene[18], we thus speculate a *LncHIFCAR*-dependent hypoxia response through a positive feedback mechanism: the hypoxia-induced *LncHIFCAR* expression and HIF-1α stabilization presumably result in an increased abundance of the *LncHIFCAR*/HIFα complex, which works in concert to further enhance the transcription of *LncHIFCAR* and the activation of HIF-1. Further study will help elucidate the possible self-regulation of the *LncHIFCAR*-HIFα axis in hypoxia. Interestingly, our data revealed that, in addition to hypoxia, *LncHIFCAR* can also enhance the expression of HIF-1 target genes under normoxic condition. Considering that in certain circumstances, the level of HIF-1α in tumour cells can remain elevated under normoxic conditions[34], it is possible that *LncHIFCAR* also plays an important role in regulating the normoxic HIF-1 signalling pathway with functional implications in tumour progression.

Overexpression of *LncHIFCAR* has been previously reported in lung[15], breast[16] and pancreatic[17] cancer tissues, whereas downregulation of *LncHIFCAR* was found in colorectal[19] and gastric[20] cancer. A recent report suggests that *LncHIFCAR/MIR31HG* favour melanoma cancer development by facilitating the binding of Polycomb-group proteins to *p16INK4A* locus, to suppress its expression[18], whereas another study showed that *LncHIFCAR* acts as an endogenous 'sponge' competing for miR-193b binding in the cytoplasm and regulate the miRNA targets[17]. Together, given its role as transcriptional

co-activator as reported here, it appears that *LncHIFCAR* possesses both nuclear and cytoplasmic functions, and probably affects tumour progression through mechanisms at least in part, depending on its subcellular localization. In addition, through association with different transcription regulator, *LncHIFCAR* can act either as a co-activator (through interaction with HIF-1α) or co-repressor (by interacting with Polycomb-group proteins). Another lncRNA *Evf2* (also known as *Dlx6os1*) was found to have dual roles in gene regulation in a similar way, depending on whether it recruits the transcription activator DLX2 or the transcription repressor MeCP2 (methyl-CpG binding-protein 2)[54,55]. These findings surely added another level of complexity to lncRNA-mediated gene regulation. It has been reported that repetitive sequence elements in lncRNAs contribute to RNA-chromatin interactions[56]. Although *LncHIFCAR* harbours repetitive elements, no obvious sequence on *LncHIFCAR* was found complementary to the target loci. At this point, it is still unclear whether the association of *LncHIFCAR* with target loci is via direct binding or mediated by additional factors. Given that several epigenetic and chromatin modifiers are required for HIF-1 activation under hypoxia[57,58], it is possible that the chromatin association of *LncHIFCAR* promotes epigenetic modification or vice versa. Further investigation will be required to understand how chromatin-associated *LncHIFCAR* promotes and stabilizes HIF-1 enrichment on the target chromatin.

One of the unmet clinical needs in managing oral cancer patients is the identification of reliable biomarkers for early detection, which is essential for timely treatment before reaching to advanced stage. However, most of protein biomarkers identified for the detection of OSCC are not clinically accessible nor harbor sufficient sensitivity for early detection. Using lncRNA as biomarker for cancer diagnosis and prognosis however has been shown to hold several advantages in light of its sensitivity, specificity, stability and easy accessibility[10]. With the following evidence, we uncovered a remarkable value of *LncHIFCAR* for oral cancer prognosis. First, high *LncHIFCAR* strongly associates with poor OS and RFS of OSCC patients. Second, *LncHIFCAR* expression strongly correlates with poorly differentiated OSCC, which is known to display early metastasis, high relapse rate and predicts poor outcomes in the patients[59,60]. Finally, the multivariate analysis identified high *LncHIFCAR* as an independent risk factor for recurrence-free patient survival. Notably, in multiple microarray and RNA-sequencing data sets where *LncHIFCAR* and HIF-1 target genes are upregulated in the cancer specimens, the levels of *HIF1A* mRNA are not significantly altered. This result echoes the controversial views of using HIF1α protein level as a prognostic marker for OSCC[3]. By contrast, *LncHIFCAR* is potentially useful to serve as a biomarker to reflect the activation status of HIF-1 signalling in OSCC. Moreover, the roles of *LncHIFCAR* associating with clinical outcomes are supported by our *in vitro* and *in vivo* experiments that demonstrate the invasion- and metastasis-promoting effects of *LncHIFCAR*. Metastasis is a complex multistep process involving early invasion, intravasation, extravasation and colonization of cancer cells. Consistent with the findings in breast[16] and pancreatic[17] cancers, we showed that *LncHIFCAR* knockdown significantly impaired the invasive ability of cultured OSCC cells and strongly compromised the distant colonization of OSCC cells in xenograft models, reinforcing the pleiotropic roles of *LncHIFCAR* in metastasis cascade. Collectively, we defined a functional link of *LncHIFCAR* to poorly differentiated tumours with metastatic potential in OSCC and uncovered a potential clinical use for *LncHIFCAR* as a prognostic marker that may help the disease management.

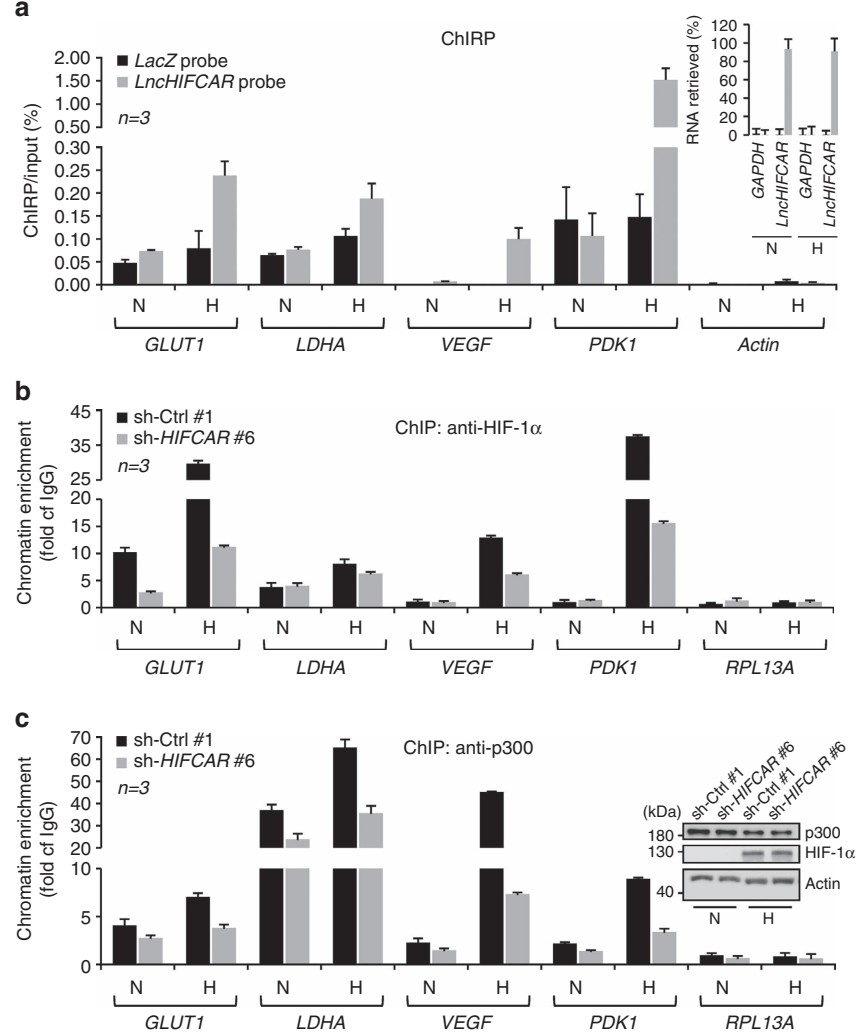

**Figure 5 | *LncHIFCAR* physically binds to the target chromatin and enhances the recruitment of HIF-1α and p300.** (**a**) ChIRP-qPCR detection of *LncHIFCAR* occupancy on the indicated target loci under normoxic and hypoxic conditions. ChIRP assays were performed with hypoxia (H)- or normoxia (N)-treated SAS cell lysate. Specific tiling biotinylated oligonucleotides complementary to *LncHIFCAR* or control *LacZ* RNA were used to pull down the RNA-associated chromatin. The inset graph shows the retrieved RNA level in the streptavidin pulled down complex, quantified by qRT–PCR. *GAPDH* mRNA was used to evaluate nonspecific binding of the biotinylated probes. The *LncHIFCAR*-associated HIF-1 target promoters were detected by qPCR. The retrieval of DNA was estimated as percentage of input chromatin, whereas *Actin* promoter served as a negative control region. (**b,c**) ChIP analysis of HIF-1α (**b**) and p300 (**c**) association with the promoter regions of HIF-1 target genes. The chromatin was prepared from vector control or *LncHIFCAR* knockdown SAS cell lines treated with normoxia (N) or hypoxia (H). ChIP was performed with anti-HIF-1α, anti-p300 or IgG and the recovered DNA was analysed by qPCR. The fold enrichment, indicated by fold of IgG, was calculated by normalizing the levels against nonspecific IgG-bound DNA. Inset, protein levels of HIF-1α and p300 in control and *LncHIFCAR* knockdown cells under normoxia (N) and hypoxia (H). Graphs show mean ± s.d. *n*, the number of independent experiments performed.

Many HIF-1 target genes control biological processes essential for cancer progression. For decades, HIF-1 and its downstream effectors such as LDHA, GLUT1 and PDK1, which mediates tumour metabolic adaptation, have been long recognized as potential targets for cancer drug due to their profound impacts in cancer progression[61–66]. However, the complexity of the HIF-1α signalling network and the requirements of targeting protein–protein interactions have made the design of HIF-1α inhibitors very challenging. Recently, targeting lncRNA has been increasingly considered as a promising therapeutic strategy for cancer treatment[10]. Because of the high expression level and the roles in HIF-1 activation in OSCC, *LncHIFCAR* holds a great potential to be developed as a specific target for OSCC therapy. To our knowledge, the present work presents the first evidence

showing that an lncRNA could modulate hypoxia signalling pathways by acting as a HIF-1α co-activator. The roles of *LncHIFCAR* in HIF-1α co-activation and the translational relevance of *LncHIFCAR* in OSCC not only provide new insight into HIF-1 activation but also reveal a potential utility of *LncHIFCAR* in prognosis and therapeutic strategy for OSCC.

## Methods

**Cell culture**. Cell lines HeLa and HEK293T (293T) were obtained from ATCC and cultured in DMEM supplemented with 10% (v/v) fetal bovine serum (FBS) (Gibco/Invitrogen). SAS cell line was obtained from JCRB cell bank and maintained in 1:1 mixture of DMEM and Ham's F12 medium (Gibco/Invitrogen) with 10% FBS. The oral cancer cell lines OC-2, OEC-M1 and HSC-3 were obtained and maintained as described previously[31]. Cell lines were not authenticated, but regularly tested for mycoplasma contamination using Venor GeM Detection Kit

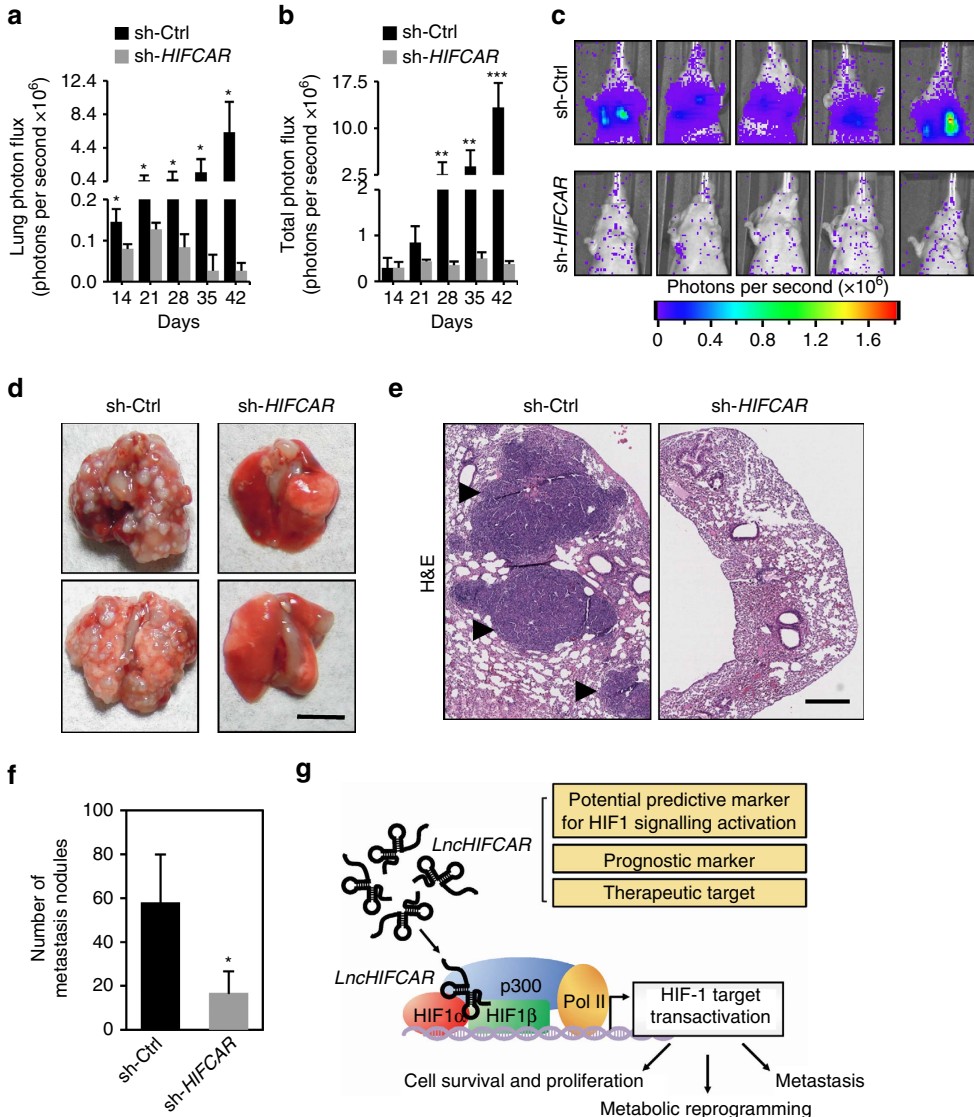

**Figure 6 | LncHIFCAR is an oncogenic driver of metastatic cascade.** SAS cells stably expressing control (sh-Ctrl) or *LncHIFCAR* knockdown (sh-*HIFCAR*) construct with firefly luciferase labelling were injected into the tail vein of nu/nu mice ($n = 10$ each group). The metastatic tumour nodules in the lungs of the animals were monitored weekly by luciferin injection and IVIS imaging. (**a,b**) The bioluminescence intensity of the lung area (**a**) and total signal (**b**) of the animals were quantified at the indicated time points. (**c**) Representative bioluminescence images of 5 mice in each group at day 42 after tail vein injections. (**d,e**) Representative images of the metastatic lung nodules (**d**; scale bar, 5 mm), and haematoxylin and eosin (H&E) staining (**e**; scale bar, 500 μm) of the lung sections from mice in each group 6 weeks after tail vein injections. The black arrowheads indicate the metastatic nodules. (**f**) Quantification of the number of pulmonary nodules in each mice group 6 weeks after tail vein injection. (**g**) A proposed model that depicts how lncRNA *LncHIFCAR* acts as a HIF-1α co-activator via its interaction with HIF-1α that contributes to the activation of HIF-1 transcriptional network associating with cancer progression. The possible utility of elevated *LncHIFCAR* expression as a prognostic biomarker and therapeutic target for OSCC is indicated. Graphs show mean ± s.d. Student's *t*-test, \**P* < 0.05, \*\**P* < 0.01 and \*\*\**P* < 0.001).

(Minerva Biolabs). None of the cell line stock used in this study is found in the database of commonly misidentified cell lines listed by ICLAC. Transfection was performed using Lipofectamine 2000 (Invitrogen) or Fugene 6 (Roche Applied Science) as recommended by the manufacturer. To generate stable *LncHIFCAR*-knockdown SAS cell lines, SAS cells transduced with *LncHIFCAR* shRNA (sh-*HIFCAR*) or empty vector (sh-Ctrl) lentiviral particles were selected by Zeocin for 3 weeks. To generate luciferase-expressing and *LncHIFCAR*-knockdown SAS cell lines for xenograft study, the SAS stable clones (sh-Ctrl 1 and sh-*HIFCAR* 6) described above were transduced with a lentiviral vector containing firefly luciferase complementary DNA, which was constructed using the ViralPower Lentiviral Gateway Expression System from Invitrogen[67], and selected by puromycin for 3 weeks. For hypoxia treatment, physical hypoxic conditions (1% oxygen) were generated by Forma Series II 3130 incubators (Thermo Scientific). Alternatively, to generate chemical-induced pseudo-hypoxic state, cells were treated with hypoxia-mimetic chemical cobalt chloride (Sigma-Aldrich) as indicated.

**Real-time qRT–PCR.** Total cellular RNA was isolated using Trizol (Invitrogen) reagent and cDNA was generated using SuperScript II first-strand synthesis system (Invitrogen). Real-time qPCR analysis was performed on the Bio-Rad iQ5 Real-Time PCR detection system (Bio-Rad) with Maxima SYBR Green qPCR Master Mix (Fermentas) or on the Rotor-Gene Q instrument (Qiagen) with QuantiFast SYBR Green PCR Kit (Qiagen). All expression levels, unless otherwise specified, were normalized against the *GAPDH* mRNA level. The primer sequences are listed in Supplementary Data 2. Real-time qPCR for miR-31 was performed using miScript PCR Starter Kit and hsa-miR-31 miScript Primer Assay according to manufacturer's instructions (Qiagen).

**Molecular cloning and siRNA transfection.** To generate sense and anti-sense biotin-labelled *LncHIFCAR* transcripts, full-length *LncHIFCAR* cDNA was amplified from HeLa nuclear RNA and cloned into pCR2.1-TOPO (Invitrogen) by TA cloning. For *in vitro* transcription of biotin-labelled *LncHIFCAR* RNA deletion

variants, the corresponding *LncHIFCAR* fragments were amplified and cloned into pGemT-Easy (Promega) by TA cloning. For ectopic overexpression of full-length *RMRP*, *LncHIFCAR* and *LncHIFCAR* (501–1,500) in human cell lines, the PCR-generated DNA fragments containing the indicated regions were inserted into pCR2.1-TOPO by TA cloning, followed by sub-cloning into the EcoRI site of pSL-MS2 vector. To knock down *LncHIFCAR* expression, DNA encoding shRNA specifically targeting *LncHIFCAR* at sequence 5′-GCTGCTGATGACGTAAAGT-3′ was cloned into pLenti4 vector. For siRNA-mediated knockdown of *LncHIFCAR*, two different siRNA oligonucleotides were synthesized and purified by by Genepharma (Suzhou, Jiangsu, China). siRNAs were transfected at a final concentration of 20 nM using Lipofectamine RNAiMAX Reagent (Invitrogen) following the manufacturer's protocol. The sequences of siRNAs are listed in Supplementary Table 2. The HIF-1α-expressing plasmid HA-HIF-1α-pcDNA3, as well as the reporter plasmid HRE-luciferase (HRE-FLuc) containing HREs fused with a firefly luciferase, were purchased from Addgene. The pRL-SV40 luciferase constitutive reporter plasmid were purchased from Promega. To express and purify GST-HIF-1α, the PCR-amplified HIF-1α fragments from HA-HIF-1α-pcDNA3 was inserted into pGemT-Easy by TA cloning, followed by sub-cloning into the NotI site of pGEX-6p-1 plasmid (GE Healthcare). To purify GST-HIF-1α deletion variants, the corresponding amplified HIF-1α fragments were cloned into the BamHI/NotI sites of pGEX-6p-1.The integrity of each construct was verified by DNA sequencing and the sequences of specific primers designed for cloning as described above are listed in Supplementary Table 3. The pLKO.1-puro plasmid-based shRNAs, including TRCN0000003808 (*HIF1A*-sh1) and TRCN0000003809 (*HIF1A*-sh2), were obtained from the National RNAi Core Facility, Institute of Molecular Biology/Genomic Research Center, Academia Sinica, Taiwan.

**cBioPortal and Oncomine database analysis.** *LncHIFCAR* expression was analysed using the Oncomine (www.oncomine.org) and cBioPortal (www.cbioportal.org) platforms. For the tumour versus normal analysis of *LncHIFCAR* (*LOC554202*) on Oncomine, the following data sets were used: Peng Head-Neck dataset (GEO: GSE25099)[25], Vasko Thyroid data set (GSE6004)[27], He Thyroid data set (GSE3467)[26], Zhao Breast data set (GSE3971)[28] and TCGA colorectal data set (TCGA Research Network; http://cancergenome.nih.gov/). The *P*-value <0.05 was considered statistically significant. For *LncHIFCAR/MIR31HG* expression analysis and co-expression network discovery, the TCGA HNSCC cohort (TCGA, Provisional)[29,30] was analysed on cBioPortal using the default options according to the instructions on the website. Pearson's and Spearman's correlations of the expression levels of 20,532 genes in 522 HNSCC cases were accessed and computed by RNA-Seq V2 RSEM data set. By default, only gene pairs with values >0.3 or <−0.3 in both measures are considered statistically significant and shown. Total 248 significantly co-expressed genes were listed, including *L1CAM* and *LDHA*.

**Cell migration and invasion assay.** A total of $3 \times 10^4$ cells were suspended in 100 μl of DMEM without FBS and seeded into the top chamber of 24-well plate-sized transwell inserts (BD Falcon, 353097) with a membrane of 8 μm pore size. The medium containing 10% FBS was placed into the lower chamber as a chemoattractant. After incubation for 24 h, the cells that did not migrate through the pores were manually removed with a cotton swab. Cells presented at the bottom of the membrane were fixed and stained with crystal violet and then counted and imaged under microscope. Cell numbers were calculated in eight random fields for each chamber and the average value was calculated. Each experiment was conducted in triplicate. Matrigel invasion assays were performed using Matrigel-coated transwell inserts with the procedure as described above.

**Sphere-formation assay.** Single-cell suspensions of SAS cells were plated (1,000 cells per well) into six-well Ultra Low Attachment plates (Corning) in serum-free DMEM/F12 culture media (Gibco/Invitrogen) supplemented with 2% B27 (Invitrogen), 20 ng ml$^{-1}$ basic fibroblast growth factor and epidermal growth factor (20 ng ml$^{-1}$, Millipore). The cells were grown in a humidified atmosphere of 95% air and 5% CO$_2$ for 15 days. Upon harvest, the spheres were counted (diameter >100 μm) with inverted phase-contrast microscopy, followed by collection for RNA extraction. For pimonidazole staining, tumour spheres grown in normoxic suspension culture were allowed to attach to 0.1% gelatin-coated cover slips for 12 h. Pimonidazole (Hypoxyprobe-1 Kit, Hypoxyprobe, Burlington, USA) was applied to the spheres for 1 h under normoxia. Intracellular pimonidazole complexes indicative of hypoxic conditions were detected by immunofluorescence microscopy using an anti-pimonidazole monoclonal antibody (1:200, Hypoxyprobe-1 Kit). Cell nuclei were counterstained by Hoechst staining, whereas necrotic cells were labelled with PI fluorescence staining. For quantification of necrotic spheres, suspended tumour spheres were collected by centrifugation for the subsequent PI fluorescence staining.

**Luciferase reporter assay.** The HIF-1α-responsive luciferase construct (pHRE-FLuc) containing HREs fused with a firefly luciferase was purchased from Addgen. For HRE luciferase assays, cells were seeded to 24-well plates at a density of $1 \times 10^5$ per well. After overnight incubation, cells were transiently co-transfected with the pHRE-FLuc reporter plasmid, empty vector, *LncHIFCAR*-expressing constructs or shRNA vector-targeting *LncHIFCAR*, as well as an internal control construct pRL-SV40 *Renilla* luciferase plasmid (Promega). Twenty-four hours post transfection, the media was replaced and the cells were exposed to 20% or 1% O$_2$ for 24 h. At 48 h post transfection, cells were lysed with passive lysis buffer and assayed for firefly and *Renilla* luciferase activities using Varioskan Flash microplate luminometer (Thermo) with the Dual-Luciferase Assay System (Promega). All the luciferase activity were normalized against the *Renilla* values and expressed as the relative fold of control group.

**Glucose uptake and lactate production assay.** The intracellular glucose and extracellular lactate were measured with the fluorescence-based glucose assay and lactate assay kits (BioVision) according to the manufacturer's instructions, respectively. Vector control and *LncHIFCAR*-knockdown SAS clones were cultured for 24 h following subsequent treatment of normoxia or hypoxia for 16 h. Intracellular glucose levels and lactate levels in the culture media were measured and presented as folds relative to the level of control cells in normoxia. All measurement were normalized by cell number.

**Nuclear and cytoplasmic fractionation.** Subcellular fractionation of protein extracts from HeLa cells was performed using NE-PER Nuclear and Cytoplasmic Extraction Reagents (Thermo Scientific) following the manufacturer's protocol. Nuclear/cytoplasmic fractionation of RNA was conducted with Nuclei EZ Lysis Buffer (Sigma), according to the manufacturer's protocol.

**Biotinylated RNA pull-down assay.** The biotinylated RNA pull-down assay was performed as described previously[68,69]. Briefly, biotin-labelled RNAs was *in vitro* transcribed with AmpliScribe T7-Flash Biotin-RNA Transcription Kit (Epicentre), treated with RNase-free DNase I and purified with an RNeasy Mini Kit (Qiagen). The lambda transcript was generated with the control plasmid provided by the Transcription Kit. To form the proper secondary structure, biotinylated RNA supplied with RNA structure buffer (10 mM Tris pH 7, 0.1 M KCl and 10 mM MgCl$_2$) was heated to 90 °C for 2 min, incubated on ice for 2 min and then shifted to room temperature (RT) for 20 min. The RNA was then mixed with hypoxic HeLa nuclear extract or purified proteins and incubated at RT for 1 h, followed by incubating with Streptavidin Mag Sepharose (GE Healthcare) at RT for 1 h. After subsequent washes, the pull-down complexes were analysed by standard western blot technique.

**Antibodies and western blot analysis.** Cells were harvested, rinsed with PBS and lysed in lysis buffer (150 mM NaCl, 50 mM Tris-HCl pH 7.4, 1 mM EDTA,1 mM MgCl$_2$, 0.5% NP-40, 1 mM Na$_3$VO$_4$, 1 mM NaF, protease inhibitors cocktail). Cell lysates were separated on SDS–polyacrylamide gel, transferred to a polyvinylidene difluoride membrane (Bio-Rad Laboratories) and immunoblotted using the following primary antibodies. Rabbit anti-HIF-1α (1:1,000, GTX127309), anti-GST (1:5,000, GTX110736), anti-lamin B2 (1:5,000, GTX109894) and anti-tubulin (1:5,000, GTX112141) antibodies, as well as mouse monoclonal anti-HIF-1α antibody (1:1,000, GTX628480), were purchased from GeneTex. Mouse mono-clonal antibodies recognizing β-actin (1:5,000, A2228) were purchased from Sigma. Mouse monoclonal HIF-1β antibody [2B10] (1:2,000, ab2771) were purchased from Abcam. Uncropped scans of the blots and gels are shown in Supplementary Fig. 19 in the Supplementary Information section.

**Purification of GST-HIF-1α.** *Escherichia coli* host BL21(DE3) harbouring the expression vector pGEX-6p-1-HIF-1α was cultured in Luria–Bertani medium with ampicillin (50 μg ml$^{-1}$) and induced by 0.3 mM IPTG (isopropyl-β-D-thioga-lactopyranoside) at 30 °C for 16 h. Affinity purification of the recombinant protein was carried out with Pierce Glutathione Superflow Agarose (Pierce) following the manufacturer's instructions.

**RNA immunoprecipitation.** RNA immunoprecipitation was performed as described previously[68,69] with the following modifications. Briefly, $2 \times 10^7$ of SAS cells treated with normoxia or hypoxia for 16 h were crosslinked with 0.3% formaldehyde in medium for 10 min at 37 °C, followed by neutralization with 125 mM glycine incubated at RT for 5 min. After two times wash with cold PBS, the cell pellets were lysed in RIPA buffer (50 mM Tris pH 7.4, 150 mM NaCl, 1 mM EDTA, 0.1% SDS, 1% NP-40 and 0.5% sodium deoxycholate, 0.5 mM dithiothreitol, RNase inhibitor and protease inhibitor cocktail), followed by sonication on ice and subsequent DNase treatment for 30 min. Immunoprecipitation were performed by incubating protein A/G precleared nuclear lysates with α-HIF-1α antibody (1:100, NB100-105, Novus Biologicals) or equivalent mouse IgG (GTX35009, GeneTex) at 4 °C overnight. The RNA/antibody complex was then precipitated by incubation with protein A/G agarose beads. After subsequent wash following the standard protocol, the RNA samples were extracted with Trizol reagent (Invitrogen) and detected by qRT–PCR.

**Immunoprecipitation.** Vector control or *LncHIFCAR* knockdown SAS cells under hypoxic conditions for 24 h were collected by centrifugation. The cell pellet was

then resuspended in lysis buffer (50 mM Tris-HCl pH 8.0, 180 mM NaCl, 1% NP-40, protease and phosphatase inhibitor cocktail) and passed through a 21-gauge syringe several times. Immunoprecipitations were performed by incubating 1 mg protein A/G precleared cell lysates with 5 µg α-HIF-1α antibody (NB100-105, Novus Biologicals) or equivalent mouse IgG (GTX35009, GeneTex) at 4 °C for 3 h. After subsequent wash, the immunoprecipated protein complex on the beads were analysed by western blotting.

**In vitro RNA-binding assay.** *In vitro*-synthesized *LncHIFCAR* RNA supplied with RNA structure buffer (10 mM Tris pH 7, 0.1 M KCl and 10 mM MgCl$_2$) was heated to 90 °C for 2 min, incubated on ice for 2 min and then shifted to RT for 20 min to form the proper secondary structure. GST fusion proteins on glutathione-Sepharose beads ($\sim$10 µl) were incubated with 2 µg *in vitro* synthesized *LncHIFCAR* RNA in 50 µl of RNA-binding buffer (50 mM Tris-HCl, pH 7.4, 100 mM KCl, 2 mM MgCl$_2$, 0.1% NP-40. 1 mM dithiothreitol and ribonuclease inhibitor) for 30 min at 4 °C. The beads were washed with RNA-binding buffer three times to remove unbound RNAs. The RNA samples retained on the beads were extracted with Trizol reagent (Invitrogen) and detected by qRT–PCR. The relative retention as to the input RNA level was calculated.

**Chromatin immunoprecipitation.** ChIP assays were performed mainly following the ChIP protocol from the University of California Davis Genome Center (http://farnham.genomecenter.ucdavis.edu). Briefly, vector control or *LncHIFCAR* knockdown SAS cell lines ($2 \times 10^7$ cells per assay) treated with normoxia or hypoxia were cross-linked with 1% formaldehyde at 37 °C for 15 min and subsequently quenched in 125 mM glycine for 5 min. The cross-linked chromatin was sonicated to generate DNA fragments averaging 200–500 bp in length. Chromatin fragments were immunoprecipitated with antibodies against HIF-1α (1:100, NB100-105, Novus Biologicals), p300 (1:100, 05-257, Millipore) or equivalent mouse IgG (GTX35009, GeneTex). The precipitated DNA were purified using the QIAquick PCR Purification Kit (Qiagen) and analysed by quantitative real-time PCR using the primers listed in Supplementary Data 2.

**Chromatin isolation by RNA purification.** ChIRP was performed using SAS cells adapting the protocols described previously[69,70] with minor modifications. Briefly, 20-mer antisense DNA probes targeting *LncHIFCAR* RNA, as well as the negative control lacZ RNA, were designed using the online probe designer at singlemoleculefish.com (http://www.singlemoleculefish.com/designer.html) as listed in Supplementary Table 4. SAS cells treated with normoxia or hypoxia for 4 h were crosslinked with 1% glutaldehyde for 10 min at RT with gentle shaking. Crosslinking was stopped with 125 mM glycine for 5 min. The cross-linked chromatin was incubated with the biotinylated DNA probes, subjected to streptavidin magnetic beads capturing and subsequent wash/elution steps essentially performed as described[70]. The eluted chromatin and RNA fragments were analysed by qPCR using the primers listed in Supplementary Data 2.

**Xenograft experiments.** Oral cancer SAS cell lines (sh-Ctrl#1 and sh-*HIFCAR*#6) used for metastasis model were all labelled with firefly luciferase as described above. In brief, $5 \times 10^5$ SAS cells (sh-Ctrl#1 and sh-*HIFCAR*#6, respectively) suspended in 0.5 ml of Matrigel (BD Bioscience) were injected into the tail veins of 6- to 8-week-old female athymic nude mice (nu/nu). Subsequently, the mice were monitored for metastases using the IVIS Lumina LT series III system (PerkinElmer) after intraperitoneal injection of luciferin once a week for 6 weeks after tail vein xenografting. Then, the lungs of the nude mice were excised post mortem for histology examination and haematoxylin–eosin staining. The experiments were not randomized. The investigators were not blinded to allocation during experiments and outcome assessment. The animal studies were approved by National Health Research Institutes Institutional Animal Care and Use Committee (approval number: NHRI-IACUC-102078) and carried out under the institutional guidelines with animal welfare standards.

**Patients and clinical samples.** The research design, study protocols and information security were approved by the Institutional Review Board of Chi-Mei Medical Center (approval number: 10312-L07). Thereby, snap-frozen primary OSCC tissues and paired non-cancerous mucosa tissues stored in liquid nitrogen were withdrawn from Chi-Mei Medical Center Tissue BioBank. Written informed consents were obtained from all participants. The clinicopathological data for the samples were presented in Supplementary Table 1.

**Statistical analysis.** All data were expressed as mean ± s.d. of three or more independent experiments. Sample sizes were selected based on experience from our previous publications. Samples were excluded only in the case where a technical error occurred during sample preparation or analysis. Statistical analyses were performed with SPSS software version 19.0 (Armonk). Differences between individual groups were analysed by two-tailed Student's *t*-test. The survival curves were calculated using the Kaplan–Meier method and the differences were assessed by a log-rank test. Statistical analyses of clinicopathological data were performed using Fisher's exact test. Univariate and multivariate Cox proportional hazards regression models were performed to identify the independent factors with a significant impact on patient survival. The HRs and 95% confidence intervals of the prognostic factors were calculated. The results were considered significant if $P < 0.05$.

**Data availability.** The TCGA and Oncomine data referenced during the study are available in a public repository from the cBioPortal for Cancer Genomics, TCGA (http://www.cbioportal.org/) and the Oncomine (www.oncomine.org) websites. For the tumour versus normal analysis of *LncHIFCAR* on Oncomine (search term: LOC554202), the following datasets were used: Peng Head-Neck dataset (Gene Expression Omnibus: GSE25099)[25], Vasko Thyroid dataset (GSE6004)[27], He Thyroid dataset (GSE3467)[26], Zhao Breast dataset (GSE3971)[28] and TCGA colorectal data set (TCGA Research Network; http://cancergenome.nih.gov/). For *LncHIFCAR* expression analysis and co-expression network discovery, the TCGA HNSCC cohort (TCGA, Provisional) was analysed on cBioPortal platform[29,30] (search term: MIR31HG). The authors declare that the data supporting the findings of this study are included within the article and its Supplementary Information files, or are available from the authors upon reasonable request.

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

## Acknowledgements

We are sincerely grateful to the following individuals at PhD Program for Translational Medicine, Taipei Medical University: Dr Chia-Ling Hsieh for providing luciferase labelling lentiviral vector; Dr Hao-Ching Wang for providing protein expression vector; Po-Yang Huang, Ju-Hui Wu, Yu-Ju Chiu and Dr Yi-Pei Lin for their technical assistance. We thank Dr Pei-Ching Chang (Institute of Microbiology and Immunology, National Yang-Ming University) for help in the initial hypoxia screening. This work was supported by grants NSC101-2917-I-564-022, NSC102-2811-B-038-008 and NSC102-2321-B-038-006 (to J.W.S.) from National Science Council of Taiwan; CLFHR10422 and CLFHR10528 (to W.F.C.) from Chi-Mei Medical Center Liouying, Taiwan; MOHW106-TDU-B-212-144001 (to Y.Y.) from Health and welfare surcharge of tobacco products; CA114575 and CA165263 (to H.J.K.) from US National Institutes of Health; NHRI03A1-MGPP18-014, NHRI04A1-MGPP15-014 and NHRI05A1-MGPP15-014 (to H.J.K.) from National Health Research Institutes of Taiwan; MOHW104-TDU-M-212-13304 (to H.J.K.) from Ministry of Health and Welfare of Taiwan; MOST102-2320-B-400-018-MY3 and MOST104-2321-B-400-009 (to H.J.K.) from Ministry of Science and Technology of Taiwan.

## Author contributions

J.-W.S. and H.-J.K. conceived and designed research. J.-W.S., W.-F.C., A.T.H.W., M.-H.W., L.-Y.W., C.-Y.C. and C.-L.H. contributed the development of new reagents, methodology and analytic tools. W.-F.C. ensured protocol integrity and collected data. J.-W.S., Y.-W.H., Y.-L.Y., L.-Y.W. and C.-L.H. performed research and collected data. J.-W.S., A.T.H.W., M.-H.W., L.-Y.W., W.-C.W., C.-L.H. and C.A.C. analysed data. J.-W.S. and H.-J.K. coordinated the project. Y.Y. provided valuable conception and support. J.-W.S., L.-Y.W. and H.-J.K. wrote the manuscript with the help from other co-authors.

**Additional information**

**Competing interests:** The authors declare no competing financial interests.

