## [Peer Review File · Nature Communications]

Reviewers' comments:

Reviewer #1 (Remarks to the Author):

Shih et al report the long non-coding RNA LncHIFCAR is induced by hypoxia and hypoxia-mimetics in cancer cell lines in vitro. Mechanistically, the authors show that LncHIFCAR augments Hif-1 transcriptional activity and target gene expression, and that it can physically interact with Hif-1 and localize at Hif-1 target gene promoters. Knockdown of LncHIFCAR reduced hypoxia-induced glucose uptake and lactate production, tumor sphere growth in vitro and lung colonization in immunocompromized mice. By analyzing publicly available databases and matched tumor/normal tissue oral squamous cell carcinomas, they also show prognostic value of high LncHIFCAR. Overall, it is a well conducted study with convincing data. The biological consequences of LncHIFCAR genetic manipulations and their impact on normoxic and hypoxic cell growth should be better refined and discussed in the context of the preclinical animal data shown in the manuscript. Major comments:

- 1) How does overexpression and knockdown of LncHIFCAR impact cell growth rates at different oxygen concentrations? It is important to know the proliferation and apoptosis rates of these cells in order to evaluate the 3D sphere formation and lung colonization data: if knockdown of LncHIFCAR slows down 2D growth then this is expected to be mirrored in tumor forming assays.
- 2) Alternatively, if cell monolayer growth rates are similar, are the knockdown cells unable to adapt to the developing hypoxia at the core of the spheres? Spheres of comparable size derived from control and knockdown cells should be stained with a hypoxic marker, such as pimonizadole or EF5, to quantify the extent of hypoxia and the extent of necrosis at the core of the spheres.
- 3) The term "hypoxia-associated promotion of stem cell properties" to describe sphere forming ability is fairly liberal and not justified by the data. Comparing the quantification of sphere formation capacity of the sh-control cells under hypoxia and normoxia (fig 2f and suppl. Fig.5) we see similar numbers for total- and >1mm spheres. A more accurate term (along the lines of anchorage-independent growth or sphere-forming ability) should be used throughout the manuscript.
- 4) In figure legends 1f-g of the Kaplan-Meier curves it is stated that "The cutoff value was defined to maximize the product of the sensitivity and specificity for each endpoint" It should be clarified how this value was defined.

Minor comments:

- 5) The brightness and contrast of the western blot panels and their corresponding uncropped scans are very high, darker images should be provided.
- 6) Line 167: correct "lactose" to "lactate"

Reviewer #2 (Remarks to the Author):

In this manuscript Shih and colleagues characterize a novel function for a long noncoding RNA that they name LncHIFCAR. They find that it is induced by hypoxia, and it is overexpressed in several cancers, included oral carcinoma. They show that LncHIFCAR contributes to the expression of hypoxia responsive genes and to the invasive phenotype of cancer cells. They propose a model where LncHIFCAR interacts with HIF-1 α , which results in the transcriptional activation of its target genes.

This is an interesting study, overall well performed and clearly written.

However, it has a major problem. The model is difficult to reconcile with the function previously reported for this lncRNA (also named MIR31HG). In the previous Nature Communications paper (Montes et al. 2015) MIR31HG was proposed to repress gene expression (instead of activate it) by interacting with PRC2 complex. In addition, it was shown to be induced upon senescence, and under those conditions be mostly localized to the cytoplasm. Senescence and hypoxia are related.

In fact in some cases hypoxia can inhibit senescence. What is the subcellular localization of the lncRNA in normoxic and hypoxic conditions? What happens to senescence markers under the experimental conditions used?

If the model proposed is correct, lncHIFCAR requires HIF1-alpha in order to induce the expression of the hypoxia genes. However, the authors show that under normoxic conditions lncHIFCAR overexpression per se is sufficient to induce hypoxia genes (fig 3a-b), although HIF1-alpha is not induced (fig 4A).

Additional points:

1. Because the initial screening is only limited to one cell type and two conditions, and only a small set of lncRNAs is analyzed, it is difficult to assess its robustness. At least a panel with a good number of positive and negative control protein-coding genes should be included.
2. Is lncHIFCAR a HIF-1 transcriptional target? What is the effect of HIF1 knockdown?
3. The RNA shown in figures 4D and 4E is not of good quality. The presence of multiple bands and smears questions the RNA pulldown data.
4. The authors show that the interaction between HIF-1alpha and lncHIFCAR is direct. What domain of the protein is able to bind RNA? How would this affect the interaction with other components of the complex?
5. In the ChIRP-PCR experiment (Fig 5a), how were the primers designed? How were the amplicons in the promoter regions selected?
6. Supp fig 2 should be more clearly explained

Response to Reviewers

We greatly appreciate both reviewers' interests in this manuscript and very helpful comments to improve the paper. We have closely followed all the reviewers' suggestions and added more than 20 pieces of new data in the figures associated with the text or supplementary sections. By following these comments, we believe that we have considerably strengthened the manuscript. Below are our point-by-point responses:

Reviewer #1:

1. *“Overall, it is a well conducted study with convincing data. The biological consequences of LncHIFCAR genetic manipulations and their impact on normoxic and hypoxic cell growth should be better refined and discussed in the context of the preclinical animal data shown in the manuscript.”*

Response:

Thanks for the positive comments and the suggestion. In the revised manuscript, we have refined the section on biological consequence of genetic manipulation by providing additional data (see below) and found that colonization in animal models correlated well with 3D hypoxic cell growth. These results are now reported in the context of preclinical animal experiments (p.10-11).

2. (Major comment 1) *“How does overexpression and knockdown of LncHIFCAR impact cell growth rates at different oxygen concentrations? It is important to know the proliferation and apoptosis rates of these cells in order to evaluate the 3D sphere formation and lung colonization data: if knockdown of LncHIFCAR slows down 2D growth then this is expected to be mirrored in tumor forming assays”*

Response:

Thanks. As suggested by the reviewer, we have analyzed the growth rate of *LncHIFCAR*-overexpressing and -knockdown of cells under normoxic and hypoxic conditions. Consistent with previous report in breast cancer cell ¹, the *LncHIFCAR*-knockdown oral cancer cells grew more slowly than the vector control (Fig. 2d) whereas *LncHIFCAR*-overexpression promoted the cell growth of SAS and

293T cells (Supplementary Fig. 6) under normoxic conditions. Most remarkably, we found much more profound differences in cell proliferation under hypoxic conditions (up to 1.7 fold, Fig. 2d) than under normoxic conditions (1.2 fold, Fig. 2d), revealing *LncHIFCAR* confers a hypoxic cell growth advantage to tumor cells which is required in several tumor progression steps, including sphere-forming ability and metastatic colonization capability. We have now included these results in Fig. 2d and Supplementary Fig. 6, and discussed these findings in lines 166-180, 355-357.

3. (Major comment 2) *“Alternatively, if cell monolayer growth rates are similar, are the knockdown cells unable to adapt to the developing hypoxia at the core of the spheres? Spheres of comparable size derived from control and knockdown cells should be stained with a hypoxic marker, such as pimonizadole or EF5, to quantify the extent of hypoxia and the extent of necrosis at the core of the spheres.”*

Response:

We have carried out this interesting experiment with the methodology suggested by the reviewer. Indeed, the dramatic difference in hypoxic 2D cell growth caused by *LncHIFCAR* knockdown as mentioned above prompted us to investigate the association of hypoxia during 3D sphere formation process. As suggested, we utilized the hypoxic marker pimonizadole to test the hypoxic status at the core of the spheres. By allowing spheres to attach to gelatin-coated cover slips, we were able to nicely stain the sphere and visualize the inside-to-outside hypoxia gradient in larger tumor sphere to validate the developing hypoxia at the core of the spheres (Fig. 3f and Supplementary Fig. S7, S8). By propidium iodide (PI) counterstaining with these attached spheres, we further identified necrotic cells in these hypoxic core region of spheres (Supplementary Fig. S8).

However, our staining procedure is not suitable for quantitation, because not every sphere could firmly attach to gelatin-coated cover slips. We have tried to counterstain sphere suspensions with PI and pimonizadole by centrifuge, but the pimonizadole signal was not robust and convincing. Therefore, we counterstained the sphere suspensions with PI and nuclei marker Hoechst instead to quantify the extent of necrosis at the core of the spheres. The increased necrotic cell death in spheres derived from *LncHIFCAR* knockdown cells as shown in Supplementary Fig. S9 suggested that these cells were unable to adapt to the developing hypoxia at the sphere core. Notably, these findings are consistent with the reduced hypoxic cell growth of *LncHIFCAR* knockdown cell as described above (Fig. 2d)

Collectively, these results underscore a crucial role of *LncHIFCAR* in hypoxia adaptation and sphere formation that may further contribute to tumor development.

We have now included these results in Fig. 3f and Supplementary Fig. S7, S8, S9 in the revised manuscript.

(Please see lines 206-216)

4. (Major comment 3) *"The term "hypoxia-associated promotion of stem cell properties" to describe sphere forming ability is fairly liberal and not justified by the data. Comparing the quantification of sphere formation capacity of the sh-control cells under hypoxia and normoxia (fig 2f and suppl. Fig.5) we see similar numbers for total- and >1mm spheres. A more accurate term (along the lines of anchorage-independent growth or sphere-forming ability) should be used throughout the manuscript."*

Response:

Good suggestions. We have now used the term "sphere-forming ability".

5. (Major comment 4) *"In figure legends 1f-g of the Kaplan-Meyer curves it is stated that "The cutoff value was defined to maximize the product of the sensitivity and specificity for each endpoint" It should be clarified how this value was defined."*

Response:

We thank reviewer for pointing out this issue. The statement in our previous version was indeed not precise. In the revised manuscript, we have clarified our definition and stated as "The cutoff point is set as the value yielding maximum sum of sensitivity and specificity for recurrence -free survival analysis", which is the exact way we determined the cutoff value (Please see lines 953-954).

6. (Minor comment 5) *"The brightness and contrast of the western blot panels and their corresponding uncropped scans are very high, darker images should be provided."*

Response:

Thanks. As suggested, all the western blot panels in the revised manuscript are

provided with darker images.

7. (Minor comment 6) *“Line 167: correct “lactose” to “lactate”*

Response:

Thanks. We have corrected this error (line 174).

Reviewer #2:

1. *“This is an interesting study, overall well performed and clearly written. However, it has a major problem. The model is difficult to reconcile with the function previously reported for this lncRNA (also named MIR31HG). In the previous Nature Communications paper (Montes et al. 2015) MIR31HG was proposed to repress gene expression (instead of activate it) by interacting with PRC2 complex. In addition, it was shown to be induced upon senescence, and under those conditions be mostly localized to the cytoplasm. Senescence and hypoxia are related. In fact in some cases hypoxia can inhibit senescence.*

Response:

Very good point. We think *LncHIFCAR*, depending on the associated factors, has both activating and repressive activities. The duality of lncRNA in transcription is an interesting topic. For instance, another lncRNA, *Evf2* (a.k.a. *Dlx6os1*), was reported to have dual roles in gene regulation, depending on whether it recruits the transcription activator DLX2 or the transcription repressor MeCP2 (methyl-CpG binding-protein 2)^{2,3}. Thus, our bias is that when *LncHIFCAR* associates with HIF-1 α , it transactivates, whereas it represses, when associating with PRC2 as described in ref. 4. In this revised manuscript, we have included the discussion about this feature of *LncHIFCAR* in lines 379-388.

2. *What is the subcellular localization of the lncRNA in normoxic and hypoxic conditions?*

Response:

As suggested by the reviewer, we analyzed the subcellular localization of the lncRNA in normoxic and hypoxic conditions with the same fractionation experimental kit as described in the previous *MIR31HG (LncHIFCAR)* report⁴. No notable changes on the subcellular localization of *LncHIFCAR* was observed (Supplementary Fig. 12) in normoxic and hypoxic conditions. A similar observation was also made in cobalt treated/untreated cells (Supplementary Fig. 12). Please see lines 246-247

3. *What happens to senescence markers under the experimental conditions used?”*

Response:

As suggested by the reviewer, we analyzed the expression of senescence markers as listed in the previous report⁴ in hypoxic and normoxic cells (the same cDNA sample in Supplementary Fig.1) by qRT-PCR with the same primer sets⁴. It was reported that some senescence markers (p16^{INK4a}, IL6, IL8, IL1a, IL1b, MMP1, ICAM, TIMP1 and CCL20) were up-regulated, whereas others (CDK4 and CDK6) are down-regulated

upon on OIS (oncogene-induced senescence) induction⁴. While we did see p16 upregulated under hypoxia conditions, the expressions of other markers vary and we were unable to draw a firm conclusion that hypoxia inhibits senescence in our experimental conditions.

4. “If the model proposed is correct, *LncHIFCAR* requires HIF1-alpha in order to induce the expression of the hypoxia genes. However, the authors show that under normoxic conditions *LncHIFCAR* overexpression per se is sufficient to induce hypoxia genes (fig 3a-b), although HIF1-alpha is not induced (fig 4A).”

Response:

We thank the reviewer for raising this interesting point which we only briefly discussed in the initial manuscript (line 331-334). It is reported that HIF-1α level in many tumor cells can remain elevated under normoxic conditions^{5,6}. Indeed, we could detect the normoxic expression of HIF-1α protein in

a series of human cancer cell lines (attached figure). Therefore, we initially speculated that, under normoxia, *LncHIFCAR* overexpression induce pseudohypoxia signature by facilitating the recruitment of HIF-1 complex, though the HIF-1α level is

relatively low.

In this revised manuscript, we investigated whether the relatively low expression of HIF-1 α under normoxia is required for *LncHIFCAR*-induced pseudohypoxia signature and HIF-1 transactivation as seen in Fig. 3a, 3b and 3h. We found that the pseudohypoxia signature and HIF-1 transactivation induced by *LncHIFCAR* overexpression under non-hypoxic conditions was reverted by HIF-1 α knockdown (Supplementary Fig. 13), which is consistent with our hypothesis. In addition, the ability of the HIF-1 α binding-deficient mutant *LncHIFCAR* (501-1500) to induce normoxic HIF-1 target genes and HIF-1 responsive luciferase reporter was severely impaired, relative to full-length *LncHIFCAR* (Fig. 4f and Supplementary Fig. 14), further supporting the critical role of normoxic HIF-1 α in *LncHIFCAR*-mediated HIF-1 target gene activation. These results are included in Supplementary Fig. S13 and S14 and described in lines 250-254, 269-275.

5. (Additional point 1) *“Because the initial screening is only limited to one cell type and two conditions, and only a small set of lncRNAs is analyzed, it is difficult to assess its robustness. At least a panel with a good number of positive and negative control protein-coding genes should be included.”*

Response:

As suggested by the reviewer, we re-analyzed the cDNA sample of our initial screening and confirmed the profound upregulation of a gene set of hypoxia-inducible protein-coding genes by qRT-PCR. In addition, 28s rRNA and ribosomal protein L13a (RPL13A) serves as hypoxia-stable housekeeping gene controls in our analysis. These results validate the robustness of our initial screening treatments and are now included as Supplementary Fig. S1 and described in lines 100-102.

6. (Additional point 2) *“Is lncHIFCAR a HIF-1 transcriptional target? What is the effect of HIF1 knockdown?”*

Response:

[REDACTED]

7. (Additional point 3) *“The RNA shown in figures 4D and 4E is not of good quality. The presence of multiple bands and smears questions the RNA pulldown data.”*

Response:

We agree with the reviewer and have optimized the experiment by adjusting the RNA preparation and the conditions of electrophoresis. In addition, we included single stranded RNA ladder marker to further distinguish the size of each RNA fragment. The results now in Fig. 4D and 4E provided solid evidence on mapping the HIF-1 α -interaction domain within *LncHIFCAR* RNA.

8. (Additional point 4) “*The authors show that the interaction between HIF-1 α and LncHIFCAR is direct. What domain of the protein is able to bind RNA? How would this affect the interaction with other components of the complex?*”

Response:

We thank Reviewer for raising this point. As suggested, we generated HIF-1 α bHLH (basic helixloop-helix; amino acids 1–80), PAS-A (Per-ARNT-Sim-A; amino acids 81–200), PAS-B (Per-ARNT-Sim-B; amino acids 201–329), and TAD (transactivation domain; amino acids 191–445) deletion mutants (Fig. 4g, bottom panel). *In vitro* pull-down assays revealed that *LncHIFCAR* strongly bound to the PAS-B domain of HIF-1 α (Fig. 4g and Supplementary Fig. 15). Because this domain is responsible for functional dimerization, we then conducted immunoprecipitation of HIF-1 α with hypoxic cell lysate to investigate the effect of *LncHIFCAR* on HIF1 complex formation. Knockdown of *LncHIFCAR* reduced the interaction between HIF-1 α and both HIF-1 β and p300 (Fig. 4h), indicating that *LncHIFCAR* could facilitate the recruitment of HIF-1 complex.

We have now included these results in Fig. 4g, 4h and Supplementary Fig. 15, and described these findings in lines 275-287.

9. (Additional point 5) “*In the ChIRP-PCR experiment (Fig 5a), how were the primers designed? How were the amplicons in the promoter regions selected?*”

Response:

We thank the reviewer for raising this point. Actually, all the primers in the ChIRP-PCR experiments are synthesized according to recent publications to amplify the promoter regions of known HIF-1 α binding sites. Due to the limitation of reference number in the main text, the associated reference is cited in the Supplementary Table 4 and Supplementary Reference in the previous version.

To emphasize the design of the primer in this revised manuscript, we add description “A set of probes complementary to *LncHIFCAR* was used to pull down the endogenous *LncHIFCAR* from normoxic or hypoxic SAS cells, and the promoter regions of known HIF-1 α binding sites were amplified and quantified by qPCR”

(please see lines 292-294) along with reference cited in the Supplementary Table 4 and Supplementary Reference 1-5.

10. (Additional point 6) "*Supp fig 2 should be more clearly explained*"

Response:

We thank Reviewer for the kindly notification. In the revised manuscript, we add more explanation about the box-plot diagrams in Supplementary fig. 3 (Supplementary fig. 2 in the previous version), including the meaning of the box, solid horizontal black line and error bars to make it more clear.

Additional Reference:

1. Shi, Y. *et al.* Long non-coding RNA Loc554202 regulates proliferation and migration in breast cancer cells. *Biochem. Biophys. Res. Commun.* **446**, 448-453 (2014).
2. Feng, J., Bi, C., Clark, B. S., Mady, R., Shah, P. & Kohtz, J. D. The Evf-2 noncoding RNA is transcribed from the Dlx-5/6 ultraconserved region and functions as a Dlx-2 transcriptional coactivator. *Genes Dev* **20**, 1470-1484 (2006).
3. Bond, A. M. *et al.* Balanced gene regulation by an embryonic brain ncRNA is critical for adult hippocampal GABA circuitry. *Nat Neurosci* **12**, 1020-1027 (2009).
4. Montes, M. *et al.* The lncRNA MIR31HG regulates p16(INK4A) expression to modulate senescence. *Nat. Commun.* **6**, 6967 (2015).
5. Kuschel, A., Simon, P. & Tug, S. Functional regulation of HIF-1alpha under normoxia--is there more than post-translational regulation? *J. Cell. Physiol.* **227**, 514-524 (2012).
6. Mills, C. N., Joshi, S. S. & Niles, R. M. Expression and function of hypoxia inducible factor-1 alpha in human melanoma under non-hypoxic conditions. *Mol. Cancer* **8**, 104 (2009).

Reviewers' comments:

Reviewer #1 (Remarks to the Author):

All comments and questions were convincingly addressed. No further revisions needed.

Reviewer #2 (Remarks to the Author):

While the authors have addressed several of my comments, some important points require further clarification.

1. It is possible that the effects observed upon HIFCAR knockdown are due to downregulation of HIF1alpha. In fact, considering loading controls of figures 4a and Sup Fig 10b, these indicate some decrease of HIF1alpha protein with shRNA#6 treatment. In addition, a very important control is missing in figures 3c and 3d, which show the mRNA levels for HIF1alpha target genes, but not of HIF1alpha itself. This important control should be shown.

2. The RNA in figure 4C still does not show a unique band, so the pulldown experiment it is not reliable.

3. To address my comment regarding the role of HIFCAR in senescence, the authors checked the levels of senescence markers in normoxic and hypoxic cells, but they didn't check them in conditions of HIFCAR knockdown. This should be done.

4. In addition, most of the experiments are done using shRNA#6 to knockdown HIFCAR. Another shRNA (or an alternative technique such as CRISPRi or CRISPR KO) should be used to show the effect on HIF1alpha response.

5. To avoid confusion in the literature, the lncRNA should be named as in the previous publication, lncRNA MIR31HG.

We are delighted that the first reviewer felt “All comments and questions were convincingly addressed”. We have closely followed the second reviewers’ suggestions and added new data in the figures associated with the text or supplementary sections. We trust that our revision has further strengthened the manuscript.

Below are our point-by-point responses to review 2

Reviewer #2:

1. *“It is possible that the effects observed upon HIFCAR knockdown are due to downregulation of HIF1alpha. In fact, considering loading controls of figures 4a and Sup Fig 10b, these indicate some decrease of HIF1alpha protein with shRNA#6 treatment. In addition, a very important control is missing in figures 3c and 3d, which show the mRNA levels for HIF1alpha target genes, but not of HIF1alpha itself. This important control should be shown.”*

Response:

Thanks. Indeed, one of the major mechanisms mediated by HIF-1 activation under hypoxia is attributed to the stabilization of HIF-1 α protein. Following the reviewer’s suggestion (this point and point 4), we re-analyzed the protein levels of HIF-1 α upon two independent siRNAs-mediated knockdown of *LncHIFCAR* as previously reported^{1,2}. As shown in our newly included data in Figure 4a and Supplementary Figure 12b, we found that neither siRNA-mediated *LncHIFCAR* knockdown had a significant impact on the protein level of HIF-1 α either under normoxic, chemical induced pseudo-hypoxic or physical hypoxic conditions, all of which were consistent with our previous findings.

In addition, the analysis of HIF-1 α mRNA was included in new figures 3c and 3d. Again, no significant effect on HIF-1 α mRNA level was found in either siRNA-transfected cells. Collectively, these results indicated a mechanism for *LncHIFCAR*-mediated HIF-1 activation independent of protein stabilization and mRNA accumulation of HIF-1 α .

2. *“The RNA in figure 4C still does not show a unique band, so the pulldown experiment it is not reliable.”*

Response:

The reviewer’s comment is well taken. In this revised manuscript, we have optimized the experiment by adjusting the conditions of electrophoresis. In addition, single stranded RNA ladder marker is also included to further distinguish the size of each RNA fragment. The results now in Fig. 4C provided solid pulldown evidence on the interaction between HIF-1 α and *LncHIFCAR* RNA, consistent with Figs. 4d and 4e.

3. *“To address my comment regarding the role of HIFCAR in senescence, the authors checked the levels of senescence markers in normoxic and hypoxic cells, but they didn’t check them in conditions of HIFCAR knockdown. This should be done.”*

Response:

We thank the reviewer for further specifying these concerns. It has been reported that *LncHIFCAR/MIR31HG* knockdown induced a senescence phenotype and an increase in p16^{INK4A} in immortalized human diploid fibroblasts¹. However, the expression of several genes belonging to the SASP (senescence associated secretory phenotype), such as IL1a, IL1b, IL8, ICAM and TIMP1, were unaffected or decreased in *LncHIFCAR/MIR31HG* knockdown fibroblasts¹. As suggested by the reviewer, we analyzed the expression of senescence markers in normoxic and hypoxic *LncHIFCAR/MIR31HG* knockdown HeLa cells by qRT-PCR with the same primer sets¹. As shown in the attached figure, *LncHIFCAR* knockdown impaired the activation of hypoxic marker (CA9 and LDHA) upon hypoxia, but had little effect on the senescence markers.

Most remarkably, we also performed β -galactosidase staining of HeLa and 293T cells under these conditions. As shown in the attached figure, we hardly found positive-staining cells in either conditions, suggesting that activation of senescence may not play a major role in our systems.

Considering the different cell types (fibroblast versus cancer cells) in the two studies, the phenotype of *LncHIFCAR* knockdown may vary depending on the cell context.

4. “In addition, most of the experiments are done using shRNA#6 to knockdown HIFCAR. Another shRNA (or an alternative technique such as CRISPRi or CRISPR KO) should be used to show the effect on HIF1alpha response.”

Response:

We thank the reviewer for raising this important point. In this revised manuscript, we have now included two independent siRNAs to knockdown *LncHIFCAR* as previously reported^{1,2}. These two siRNAs were used to further validate the effect of *LncHIFCAR* on HIF-1 α response in Figs. 2b, 2c, 2d, 2e, 2f, 2g, 3c, 3d and 3i. Notably, these results are consistent with our previous findings in shRNA knockdown cell lines, corroborating the oncogenic role of *LncHIFCAR* as a HIF-1 α co-activator that regulates the HIF-1 transcriptional network. In addition, the previous results in shRNA knockdown cell lines were moved to Supplementary Fig. 6, 7 and 11 to support our findings.

5. “To avoid confusion in the literature, the lncRNA should be named as in the previous publication, lncRNA MIR31HG.”

Response:

We thank the reviewer for raising this point. To avoid confusion, in this manuscript, we will use “*LncHIFCAR/MIR31HG*”.

Notably, the name of *MIR31HG* (miR-31 host gene) is hard to distinguish between the *pre-spliced form* (*MIR31HG* primary RNA, which contains miR-31) and the *spiced form*, *mature MIR31HG* RNA (lacking miR-31). As illustrated in the attached figure, unlike the case of *MIR155HG*, the miR-31 portion is spliced out and no longer contained in the sequence of mature *MIR31HG* lncRNA. Given that its function is independent of miR-31, we therefore suggest the name *LncHIFCAR* (long noncoding HIF-1 α co-activating RNA) to specify this RNA species. For this first publication, we agree with the reviewer that we should refer to it as lncRNA *LncHIFCAR/MIR31HG* in the title and abstract to

help readers link with previous literature. We have also described their relationship in the manuscript (lines 84-88). We hope that this would help readers to appreciate this miR-31-independent RNA species of novel functions.

Additional Reference:

1. Montes, M. *et al.* The lncRNA MIR31HG regulates p16(INK4A) expression to modulate senescence. *Nat. Commun.* **6**, 6967 (2015).
2. Yang, H. *et al.* Long noncoding RNA MIR31HG exhibits oncogenic property in pancreatic ductal adenocarcinoma and is negatively regulated by miR-193b. *Oncogene* **35**, 3647-3657 (2016).

REVIEWERS' COMMENTS:

Reviewer #2 (Remarks to the Author):

All my comments have been addressed